# Assembly of infectious Kaposi's sarcoma-associated herpesvirus progeny requires formation of a pORF19 pentamer

Peter Naniima[1,2☉], Eleonora Naimo[1,2☉], Sandra Koch[1,2], Ute Curth[3], Khaled R. Alkharsah[4], Luisa J. Ströh[1], Anne Binz[1,5], Jan-Marc Beneke[6], Benjamin Vollmer[7], Heike Böning[1], Eva Maria Borst[1], Prashant Desai[8], Jens Bohne[1], Martin Messerle[1,2,5], Rudolf Bauerfeind[9], Pierre Legrand[10], Beate Sodeik[1,2,5], Thomas F. Schulz[1,2,5], Thomas Krey[1,2,5,6,11]*

1 Institute of Virology, Hannover Medical School, Hannover, Germany, 2 German Center for Infection Research (DZIF), partner sites Hannover-Braunschweig and Hamburg-Lübeck-Borstel-Riems, Braunschweig, Germany, 3 Institute for Biophysical Chemistry, Hannover Medical School, Hannover, Germany, 4 Department of Microbiology, College of Medicine, Imam Abdulrahman Bin Faisal University (IAU), Dammam, Saudi Arabia, 5 Excellence Cluster 2155 RESIST, Hannover Medical School, Hannover, Germany, 6 Center of Structural and Cell Biology in Medicine, Institute of Biochemistry, University of Luebeck, Luebeck, Germany, 7 Centre for Structural Systems Biology, Leibniz-Institut für Experimentelle Virologie (HPI), Hamburg, Germany, 8 Department of Oncology, The Sidney Kimmel Comprehensive Cancer Center at Johns Hopkins, The Johns Hopkins University, Baltimore, Maryland, United States of America, 9 Research Core Unit Laser Microscopy, Hannover Medical School, Hannover, Germany, 10 Synchrotron SOLEIL, L'Orme des Merisiers, Gif-sur-Yvette, France, 11 Centre for Structural Systems Biology (CSSB), Hamburg, Germany

☉ These authors contributed equally to this work.
* krey@biochem.uni-luebeck.de

**Data Availability Statement:** The atomic coordinates and structure factors for three structures have been deposited in the protein database (PDB) under the accession numbers

## Abstract

Herpesviruses cause severe diseases particularly in immunocompromised patients. Both genome packaging and release from the capsid require a unique portal channel occupying one of the 12 capsid vertices. Here, we report the 2.6 Å crystal structure of the pentameric pORF19 of the γ-herpesvirus Kaposi's sarcoma-associated herpesvirus (KSHV) resembling the portal cap that seals this portal channel. We also present the structure of its β-herpesviral ortholog, revealing a striking structural similarity to its α- and γ-herpesviral counterparts despite apparent differences in capsid association. We demonstrate pORF19 pentamer formation in solution and provide insights into how pentamerization is triggered in infected cells. Mutagenesis in its lateral interfaces blocked pORF19 pentamerization and severely affected KSHV capsid assembly and production of infectious progeny. Our results pave the way to better understand the role of pORF19 in capsid assembly and identify a potential novel drug target for the treatment of herpesvirus-induced diseases.

## Introduction

Herpesviruses are a large group of double-stranded DNA viruses that establish latent, lifelong infections in both vertebrate and nonvertebrate species. They are classified into 3 subfamilies

7NXP (HCMV pUL77CTD), 7NXQ (KSHV pORF19KCTD) and 7NXR (MuHV-68 pORF19MCTD). All other relevant data are within the paper and its Supporting Information files.

**Funding:** This work was supported by funding to TK by the German Center of Infection Research (DZIF, www.dzif.de/en), by the Deutsche Forschungsgemeinschaft (DFG, German Research Foundation, www.dfg.de) under Germany's Excellence Strategy – EXC 2155 – Projektnummer 390874280, by the Deutsche Forschungsgemeinschaft (DFG) - Projektnummer 158989968 - SFB 900, project B10 and by funding to EMB and TK by the Deutsche Forschungsgemeinschaft (DFG) - Projektnummer 441233738. The funders had no role in study design, data collection and analysis, decision to publish, or preparation of the manuscript.

**Competing interests:** The authors have declared that no competing interests exist.

**Abbreviations:** ATCC, American Type Culture Collection; AUC, analytical ultracentrifugation; CATC, capsid-associated tegument complex; cryo-EM, cryo-electron microscopy; CTD, carboxyl-terminal domain; CVSC, capsid vertex–specific component; DAPI, 4′,6-diamidino-2-phenylindole; ECL, enhanced chemiluminescence; FACS, fluorescence-activated cell sorting; FBS, fetal bovine serum; HAT, histone acetyltransferase; HCMV, human cytomegalovirus; HDAC, histone deacetylase; HRP, horseradish peroxidase; HSV, herpes simplex virus; KSHV, Kaposi's sarcoma-associated herpesvirus; LTR, long-terminal repeat; MCMV, murine cytomegalovirus; MCP, major capsid protein; MuHV-68, murid gammaherpesvirus 68; PFA, paraformaldehyde; PRV, pseudorabies virus; PVAT, portal vertex–associated tegument; RFP, red fluorescent protein; RT, room temperature; SB, sodium butyrate; SCP, small capsid protein; SEC, size exclusion chromatography; wt, wild-type.

(α-, β-, and γ-*Herpesvirinae*) based on their genetic relationship and their biological properties [1]. Nine of them infect humans and cause a number of diseases ranging from minor disorders to fatal clinical manifestations [2]. Herpesvirus virions share a common 3D organization with the double-stranded DNA genome packaged into a pseudo-icosahedral capsid (triangulation number T = 16), which is embedded into a proteinaceous tegument layer and the viral envelope [3]. The capsid consists of 150 hexon capsomers forming the capsid faces, 11 penton capsomers located at 11 capsid vertices, and 1 DNA portal located at the 12th vertex. The hexons are composed of 6 copies of the major capsid protein (MCP; pORF25 in Kaposi's sarcoma-associated herpesvirus (KSHV); for a comparative nomenclature of selected herpesvirus capsid proteins, see S1 Table) and the small capsid protein (SCP; pORF65 in KSHV), while the pentons contain 5 copies of MCP and in some herpesviruses also SCP. Heterotrimeric protein complexes, the triplexes, cross-link the capsomers at their bases. The portal consists of a dodecameric ring of the portal protein [4–8] (pORF43 in KSHV) that facilitates regulated packaging of the viral genomes during assembly and their release from incoming capsids into the nucleoplasm [8,9].

Even though the general capsid architecture is conserved across the herpesvirus family, capsid assemblies of α-, β-, and γ-herpesviruses also show some distinct structural features. Cryo-electron microscopy (cryo-EM) analyses identified 2 copies of the portal capping protein pUL25 of the α-herpesvirus herpes simplex virus (HSV) and its ortholog pORF19 of the γ-herpesvirus KSHV, respectively, around the penton vertices as part of a protein complex termed "capsid vertex–specific component" (CVSC; [10–12]) or "capsid-associated tegument complex" (CATC; [4,13–15]; reviewed in [16]). Structural information about pUL25 and its orthologs is available to date in form of a crystal structure of the carboxyl-terminal globular domain of pUL25 (PDB 2F5U; [17]) that is arranged around the penton, as well as atomic resolution cryo-EM reconstructions of the N-terminal part of α- and γ-herpesvirus pUL25 orthologs that connects the respective carboxyl-terminal domains (CTDs) via the CVSC helical bundle and the second major component of the CVSC (pUL17 for HSV and its positional homologs pUL93 (HCMV) and pORF32 (KSHV)) to the triplex proteins (S1 Fig).

CVSC components of α- and γ-herpesviruses have different occupancies [4], explaining observed variations in the CVSC density, but are both associated with the penton vertices in a similar manner. In contrast to α- and γ-herpesviral capsids, recent cryo-EM reconstructions of the human and murine cytomegaloviruses (HCMV and MCMV), which belong to the β-*Herpesvirinae*, did not reveal any densities around the penton vertices that could correspond to a β-herpesviral CVSC. Instead, the β-herpesvirus–specific phosphoprotein pp150 forms a dense capsid-binding layer positioned around pentons and hexons [18,19]. Similarly, all prospective CVSC binding sites near the pentonal vertices on triplexes of the human herpesvirus 6B are occupied by a tetramer of the β-herpesvirus–specific pU11 protein [20], indicating that the spatial organization of the CVSC around the pentons is not conserved across herpesvirus subfamilies.

In spite of this apparent difference in capsid association, the α-herpesviral CVSC components (pUL17 and pUL25) and their β-herpesviral orthologs (pUL93 and pUL77) serve similar functions. In both α- and β-herpesviruses, they form a complex, and they are essential for packaging of viral DNA genomes; in addition, the α-herpesviral CVSC proteins stabilize capsids after genome packaging and facilitating nuclear egress of DNA-filled capsids (reviewed in [16]). The α-herpesviral CVSC stably interacts with the carboxyl-terminal part of the inner tegument protein pUL36 and connects the tip of the penton with the capsid triplexes [15,21]. pUL25 is not required for cleavage of concatemeric viral DNA [10], but for stabilization of the capsid against the internal pressure of the highly charged DNA genomes after packaging [22–25] (reviewed in [16]). Of note, pUL25 has been implicated in DNA cleavage during packaging

termination and direct interaction with DNA [26], and a recent cryo-EM study demonstrates that the DNA genome terminates in close proximity to the electron density of the portal cap likely consisting of pUL25 [5]. In HCMV, pUL77 interacts with the DNA packaging proteins pUL56 and pUL89 [27], also known as the terminase complex, although another study did not find evidence for such a direct interaction [28]. In addition, pUL77 was described to bind DNA and interact with the portal protein pUL104 [29], suggesting that pUL77, like its HSV-1 ortholog pUL25, is involved in DNA packaging. However, in contrast to its HSV-1 ortholog pUL25, HCMV pUL77 is also required for DNA cleavage [28]. HSV-1 mutants lacking the second CVSC component pUL17 are not able to cleave concatemeric DNA and therefore can also not package unit-length genomes [30]. A recent report identified pUL17 as one of the key viral factors for engaging the terminase complex [31]. Similarly, HCMV pUL93 also plays a crucial role for cleavage of the viral DNA [28,32]. The CVSC is also involved in nuclear egress of mature capsids, as HSV-1 pUL25 directly binds to the nuclear egress complex [33], and capsids of a pseudorabies virus (PRV) mutant lacking functional pUL25 accumulated at the inner nuclear membrane [34].

Recent technical advances in cryo-EM reconstructions of herpesviral capsids applying focused classification or symmetry-relaxed image processing strategies facilitated novel insights into the architecture of the portal vertex [4–6]. The portal is a dodecamer [8,9] and capped by a star-shaped pentameric density in line with the tail-like structure previously identified at the portal vertex [35]. This density was proposed to consist of pentameric layers of the CTD of pUL25 or pORF19 in HSV-1 and KSHV, respectively [4–6], in agreement with its role to prevent leakage of encapsidated viral genomes from capsids (for a schematic comparison of penton and portal vertex, see S1 Fig). Unfortunately, the lower resolution in the portal cap region of the reported cryo-EM maps did not allow building of an atomic model, leaving important mechanistic questions on the assembly of the portal cap as well as protein–protein and protein–DNA interactions during DNA encapsidation unanswered.

Here, we report the crystal structures of the CTDs of pUL25 orthologs from 3 members of the β- and γ-herpesviruses (HCMV, KSHV, and murid gammaherpesvirus 68 (MuHV-68)), highlighting their structural conservation across the herpesvirus subfamilies. Moreover, we determined the crystal structure of an intact pentameric assembly of the KSHV pORF19 CTD that displays a striking similarity to the reported portal cap density [4]. Pentamerization of this CTD could be triggered in solution, providing new insights into the dynamics of oligomerization during capsid assembly. Structure-based mutagenesis of the lateral pentameric pORF19 interface severely impaired capsid assembly and virion release, demonstrating that the formation of the pentameric portal cap is a key step of the herpesviral replication cycle. We thereby identify a novel promising target allowing for the structure-based design of antiherpesviral inhibitors.

## Results

### Crystal structures of pORF19 and its orthologs reveal a conserved 3D-fold

Cryo-EM analyses of herpesviral capsids show that the HSV-1 minor capsid protein pUL25 and its KSHV ortholog pORF19 are arranged around the penton as part of the CVSC (S1 Fig) [4,5,11–13,15]. HCMV pUL77 is considered the ortholog of pUL25 [28], but despite biochemical evidence indicating its association with capsids [28], the analogous positions around the penton vertices are occupied by a dense layer of the β-herpesvirus–specific phosphoprotein pp150 [19,36]. To complement the available structural information on pUL25 ([17]; PDB 2F5U), we set out to crystallize the CTDs of pUL25 orthologs from β- and γ-herpesviruses. We aligned the amino acid sequences of the HCMV, KSHV, and MuHV-68 orthologs

with the HSV-1 pUL25 sequence and expressed CTDs (pUL77$_{CTD}$ (HCMV), pORF19$_{KCTD}$ (KSHV), and pORF19$_{MCTD}$ (MuHV-68)) corresponding to the crystallized pUL25 carboxyl-terminal globular domain [17]. Size exclusion chromatography (SEC) revealed that the corresponding proteins were monomeric in solution as estimated by their elution volumes (S2 Fig), and we obtained diffraction quality crystals for all of them. Details of crystallization and structure determination are described in Materials and methods, and the crystallographic statistics are listed in S2 Table.

The experimental electron density maps allowed tracing of 423 out of 463, 394 out of 427, and 385 out of 413 amino acids for pUL77$_{CTD}$, pORF19$_{KCTD}$, and pORF19$_{MCTD}$, respectively (see Materials and methods; Fig 1A–1D), with internal breaks at several internal disordered loops (S3 Table). The overall structures of the HSV-1, HCMV, KSHV, and MuHV-68 orthologs are well conserved (Fig 1A–1D) as reflected by high z scores resulting from structural comparisons using the Dali server (S4 Table) [37] and in spite of a low amino acid identity within the "twilight zone" [38] (approximately 20% between the crystallized fragments of HSV-1 pUL25 and MuHV-68 pORF19; illustrated in the structural alignment in S3 Fig). This structural similarity was anticipated for the α- and γ-herpesviral orthologs based on their similar CVSC organization and shape on the respective capsids [4,5,11–13,15], but unanticipated for the β-herpesviral HCMV pUL77, for which no prior structural information was available to date. The striking structural similarity between pUL77 and its α- and γ-herpesvirus orthologs suggests a conserved function, in line with the essential role described for the CVSC in genome encapsidation and capsid maturation [23,28].

Maintaining the same function throughout virus evolution, such as the interaction with a viral or host protein, could require the conservation of nonadjacent residues to form a continuous surface patch on all herpesvirus orthologs. To test this hypothesis, we structurally aligned all 4 CTDs and identified a number of conserved residues (yellow in Fig 1E and red fonts boxed in black in S3 Fig). These residues are distributed throughout the entire protein with the majority of them located in the interior, consistent with their importance for the structural integrity of the protein core. The only larger conserved patch exposed on the molecule surface covers an area of 590.31 Å$^2$ (orange in Fig 1E) and is located in the immediate proximity of a surface loop of variable size in the different orthologs (residues 417 to 425 for pUL25$_{CTD}$, residues 472 to 492 for pUL77$_{CTD}$, residues 400 to 407 for pORF19$_{KCTD}$, and residues 366 to 383 for pORF19$_{MCTD}$). This loop was disordered in all structures (N-terminal and carboxyl-terminal residues highlighted by black arrows in Fig 1E and S3 Table) and therefore likely masks the conserved patch in the native protein, at least partially. In conclusion, the divergent protein surfaces across the 4 orthologs do not indicate any conserved protein–protein interaction interface.

The crystal structure of the HSV-1 pUL25 CTD displays a remarkable charge distribution with a number of positively charged residues on one face of the molecule (blue in S4A Fig) and a cluster of negatively charged residues on the opposite face ([17]; red in S4B Fig). Although a possible role of these positively charged patches in DNA interaction has been discussed [17], the implications of this charge distribution remain elusive. A similar but more pronounced polarization in the electrostatic potential was observed in pUL77$_{CTD}$, whereas the corresponding surfaces of the γ-herpesviral orthologs did not reveal any accumulation of charged residues (S4 Fig).

## Crystal structure of KSHV pORF19 reveals a stable pentameric assembly

Similar to HSV-1 pUL25$_{CTD}$ [17], pUL77$_{CTD}$ and pORF19$_{MCTD}$ crystallized as a monomer. By contrast, pORF19$_{KCTD}$ assembled as pentamers in 3 independent crystal forms with either a

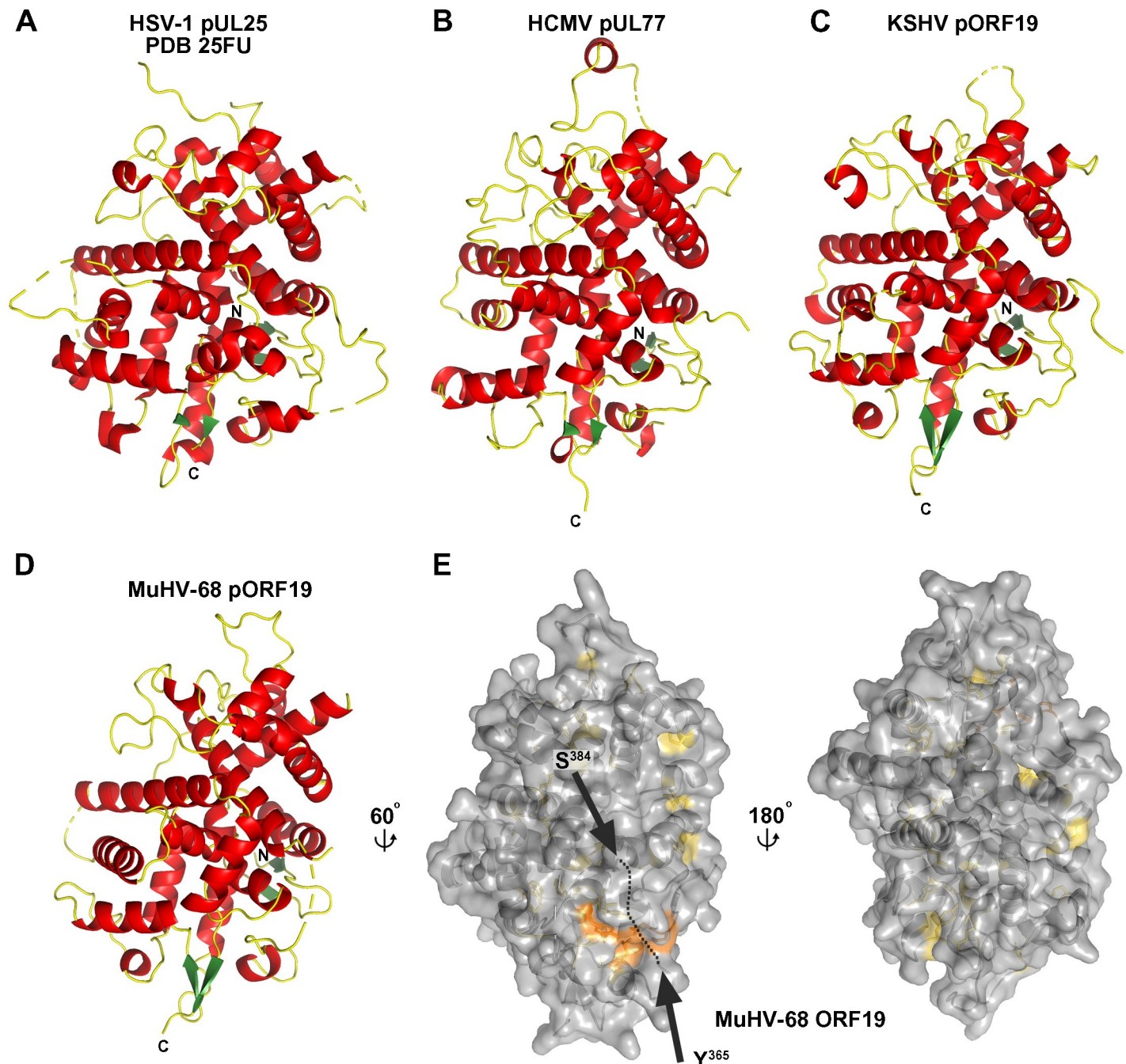

**Fig 1. Crystal structure of the minor KSHV capsid protein pORF19 and its orthologs. (A–D)** Cartoon representation of CTDs derived from HSV-1 pUL25 (PDB 2F5U; 18), HCMV pUL77, KSHV pORF19, and MuHV-68 pORF19 colored according to secondary structure with α-helices in red, β-strands in green, and loop regions in yellow. Disordered regions are shown as dashed tubes, N-terminal and carboxyl terminus are indicated. **(E)** View on the MuHV-68 pORF19 CTD (pORF19$_{MCTD}$) shown in surface representation with residues conserved across all 4 orthologs colored in yellow. A larger conserved patch consisting of 4 residues distant in primary sequence (colored in orange) is in close proximity to the loop connecting Y$_{365}$ and S$_{384}$ (black arrows), which is disordered in the crystal structures of all orthologs, suggesting that the conserved patch is likely buried in the native protein. CTD, carboxyl-terminal domain; HCMV, human cytomegalovirus; HSV, herpes simplex virus; KSHV, Kaposi's sarcoma-associated herpesvirus; MuHV-68, murid gammaherpesvirus 68.

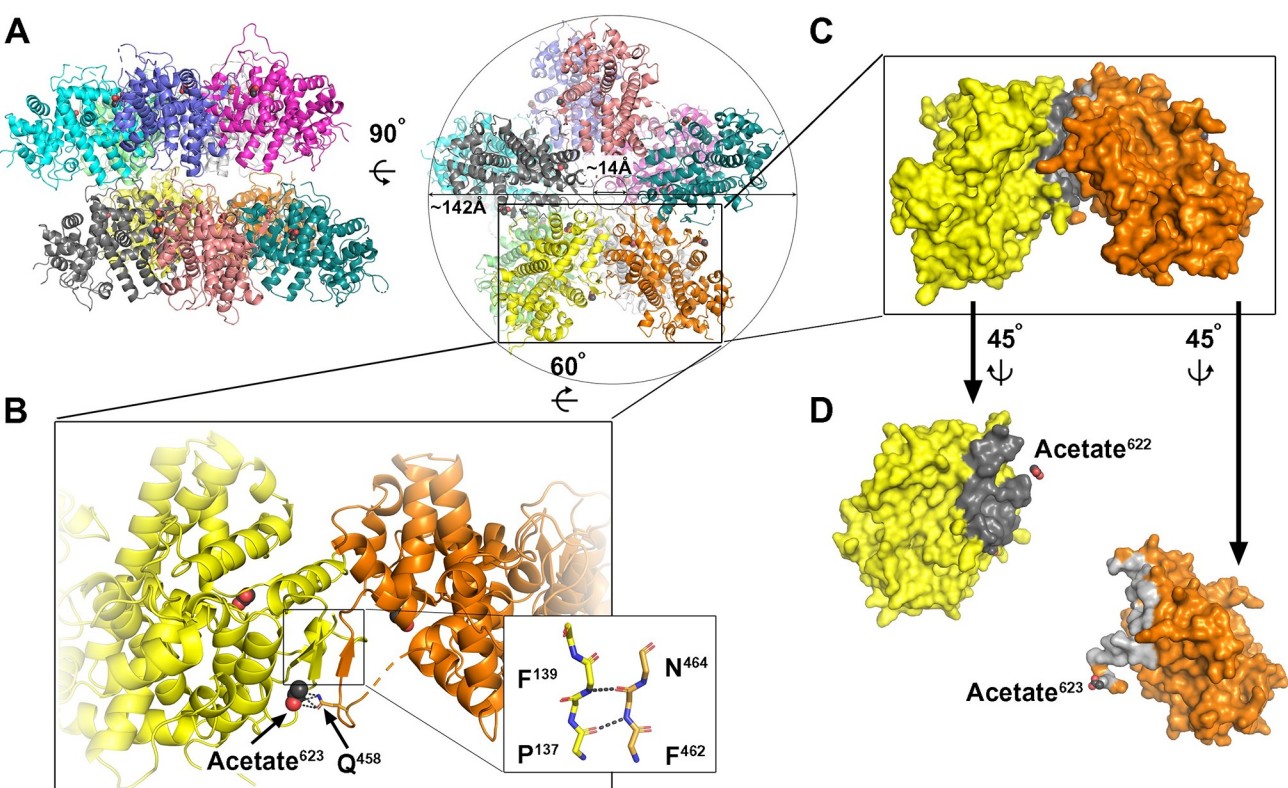

**Fig 2. Pentameric arrangement of pORF19$_{KCTD}$.** (A) Cartoon representation of pORF19$_{KCTD}$ in its decameric form as observed in the P212121 spacegroup in side view (left panel) and top view (right panel). Individual protomers are colored differently. (B) Close-up view of the lateral interface region with the 2 observed acetate ions shown as spheres and the polar interactions of acetate$^{623}$ with the side chain of Q$^{458}$ stabilizing the pentameric interface shown as dotted lines. The inset highlights the inter-protomer main chain hydrogen bonds between the 2 β-strands from adjacent protomers likely conferring extra stability to the pentamer. (C) Surface representation of the same view as in B with the interface highlighted in light and dark gray on the 2 protomers, respectively. (D) Close-up view on the interface region highlighting the position of the 2 acetate ions (shown as in B).

single pentamer or 2 pentamers per asymmetric unit (Fig 2, S5 Fig). A structural comparison of the 5 available individual pentameric rings revealed a striking similarity with root mean squared deviations between Cα-atoms of <1.6 Å. The pentamer has an outer diameter of approximately 142 Å, a central cavity of approximately 14 Å that is slightly narrower than the diameter of a double-stranded DNA helix (approximately 20 Å) and lateral interactions that bury an area of approximately 1,049 Å$^2$ (±44.5 Å$^2$) on one protomer and approximately 1,147 Å$^2$ (±53.1 Å$^2$) on the adjacent protomer (Fig 2D). An extensive interaction network consisting of hydrophobic interactions, hydrogen bonds, and a salt bridge stabilizes this interface, and its calculated total binding energy amounts to approximately 19.2 kcal/mol (±0.9 kcal/mol). The solvent accessible surface area buried by the lateral interaction and the interacting interfaces resemble those of antibody–antigen complexes [39]. Together with the unrelated crystal packing observed in the different crystal forms, these features suggest that this pentameric assembly reflects a genuine oligomeric state of pORF19$_{KCTD}$.

The protomers also interact via an intermolecular β-sheet formed by a surface loop (residues 454 to 467) comprising a short β-strand (residues 462 to 464), which runs parallel to an N-terminal 2-stranded β-sheet of the adjacent protomer (Fig 2B). These inter-protomer main chain β-interactions are likely to confer extra stability to the pentamer. Of note, the surface loop comprising this short β-strand was proposed to insert into a groove between pORF25 and pORF65 at the penton vertex and thereby constitute the main interface for penton association

of the carboxyl-terminal KSHV pORF19 domain [4]. Our crystal structure also reveals that each protomer binds 2 acetate ions via 4 intra-protomeric hydrogen bonds (acetate[622]) and 3 salt bridges (acetate[623]), respectively. One of these acetate ions (acetate[623]) connects 2 adjacent protomers via polar interactions with the side chain of Gln[458] of the adjacent protomer, thereby further stabilizing the pentameric interactions (Fig 2B).

The spatial arrangement of the crystallized pentamer is reminiscent of the pentameric density capping the portal channel termed portal vertex–associated tegument (PVAT; (Fig 3A, EMDB 20431; [4]), which differs considerably from the penton-associated CVSC on KSHV capsids (Fig 3B, EMDB 6038; [13]). Following recent cryo-EM analyses of HSV-1 and KSHV capsids, this portal cap has been proposed to consist of a pentameric assembly of the pUL25 or pORF19 CTDs [4–6]. Fitting of the atomic coordinates of our pORF19$_{KCTD}$ pentamer into the density of the KSHV portal cap [4] revealed a high quality of fit (cross-correlation coefficient 0.78; Fig 3A bottom panel), supporting the notion that the pentameric arrangement of pORF19$_{KCTD}$ likely reflects the spatial arrangement of pORF19 within the portal cap of infectious virions. The 3 independent crystal packing environments obtained for the pentameric pORF19$_{KCTD}$ (S5 Fig) suggest that one pentamer serves as building block, which in HSV-1 could subsequently adopt a "stacked ring" architecture [5,6] to form a double layer for the portal cap.

Due to the limited resolution of the portal cap density, we were unable to unambiguously determine the orientation of our pentameric pORF19 atomic model by fitting into the PVAT density, although the cross-correlation coefficients suggested a preference for the funnel shape to be oriented toward the portal. As described above the molecular surface of the individual pORF19$_{KCTD}$ protomer does not exhibit any apparent clustering of positive or negative electrostatic charges into patches—unlike the large number of positively charged residues on the surface of the HSV-1 pUL25$_{CTD}$ [17]. By contrast, the pORF19K$_{CTD}$ pentamer displays a striking enrichment of positively charged patches, lining the funnel interior (Fig 3C). This remarkable charge distribution is in line with a direct electrostatic interaction between the portal cap and the negatively charged phosphate backbone of the viral DNA genome and is also supported by cryo-EM analysis of HSV-1 capsids showing the terminal DNA running through the portal vertex [5]. Only one of the positively charged residues lining the funnel interior is conserved across herpesviruses, suggesting that other residues in this region take over that function in pORF19 orthologs. With the pentamer funnel oriented toward the portal the N-termini of the pORF19$_{KCTD}$ protomers point toward the capsid and connect to linker regions comprising 22 residues that can neither be accounted for by our crystal structure nor by the recent asymmetric (C5) reconstruction of the KSHV portal vertex [4]. This linker region is visible at lower thresholds in the reconstruction and likely involved in protein–protein interactions anchoring the cap on the portal vertex, since the "tentacle helices" emanating from the clip region of the portal toward the portal cap do not extend to the pentameric cap density [18].

## Structure-based mutagenesis in the lateral pORF19KCTD interface blocks pentamerization

To better understand pentamer formation, we analyzed the oligomerization state of pORF19$_{KCTD}$ in solution. SEC analysis in low salt buffer (SEC buffer, 10 mM Tris pH 7.5, 50 mM NaCl) did not provide evidence for the presence of a pentamer in solution (S2B Fig). As the 3 crystallization conditions contained approximately 1.7 M lithium acetate, we studied the behavior of pORF19$_{KCTD}$ by analytical ultracentrifugation (AUC) in sedimentation velocity experiments in a low and a high ionic strength buffer, containing 50 mM NaCl (SEC buffer) and 0.5 M lithium acetate, respectively. A sedimentation coefficient ($s_{20, w}$) of 3.4 S in SEC

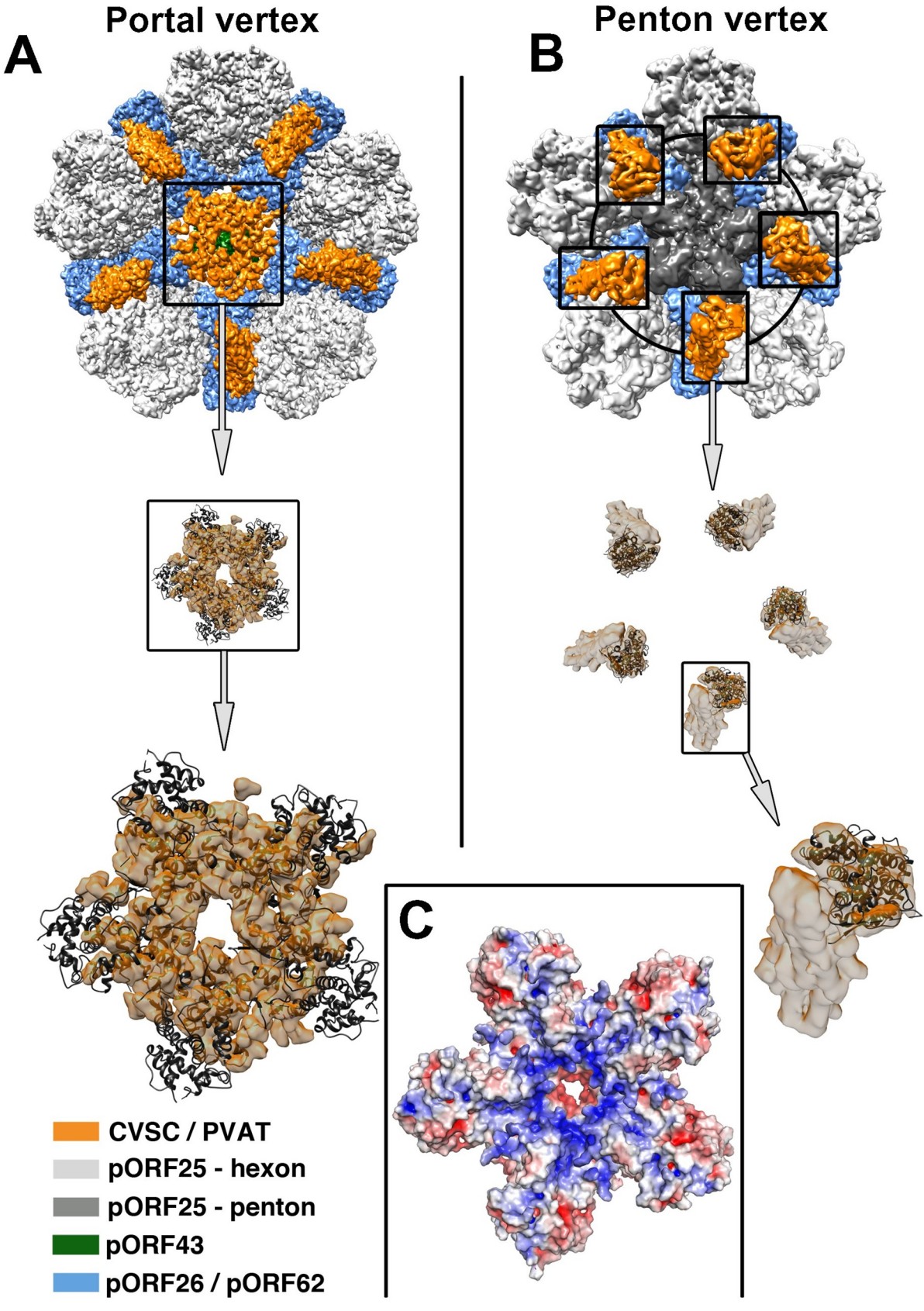

**Portal vertex**

**Penton vertex**

**A**

**B**

**C**

CVSC / PVAT
pORF25 - hexon
pORF25 - penton
pORF43
pORF26 / pORF62

**Fig 3. Arrangement of the pORF19$_{KCTD}$ pentamer on the capsid. (A)** Cartoon representation of the pentameric pORF19$_{KCTD}$ ring fitted into the portal cap electron density of the asymmetric (C1) KSHV portal vertex reconstruction (EMDB 20431; [4]) and **(B)** the pORF19$_{KCTD}$ protomers (colored in dark gray) fitted into the CVSC density of a symmetric KSHV penton vertex reconstruction (EMD 6038; [13]). Capsid proteins are colored as in S1 Fig. The close-up view (A, bottom panel) underlines the high quality of the fit (cross-correlation coefficient of 0.78 calculated by Chimera [40]). **(C)** The electrostatic potential on the molecular surface of the pentameric pORF19$_{KCTD}$ ring, represented ramp-colored from red (negative) to blue (positive) through white (neutral), calculated using APBS and contoured on a scale ranging from −5 to +5 kT/e, reveals an accumulation of positive charges in or around the funnel at the center of the pentamer, suggesting an electrostatic interaction with the phosphate backbone of the viral genome terminus. CVSC, capsid vertex–specific component; KSHV, Kaposi's sarcoma-associated herpesvirus; PVAT, portal vertex–associated tegument.

buffer and a molecular mass of 46 kDa were obtained from sedimentation coefficient and diffusion broadening of the sedimenting boundary by c(s) analysis (see Materials and methods). These data match the theoretical molecular mass of 47.3 kDa calculated for pORF19$_{KCTD}$ and support the notion that pORF19$_{KCTD}$ is a monomer under these conditions (Fig 4A left panel). At a protein concentration of 63 μM and in the presence of 0.5 M lithium acetate a second peak with a higher s-value appeared in the c(s) distribution, the fraction and s-value of which gradually increased in a concentration dependent manner (Fig 4A right panel), resulting in an s-value of 6.9 S at the highest protein concentration used (254 μM). This is a clear indication of pORF19$_{KCTD}$ oligomerization under these conditions. An increase in s-value with increasing protein concentration as described here for pORF19$_{KCTD}$ is typical for oligomerization reactions that are fast on the time scale of sedimentation, since the oligomer fraction grows and monomers and oligomers readily interconvert during centrifugation [41]. As long as a monomer fraction is still present, the observed s-value will be between that of the pure highest order oligomer and the monomer, and the precise oligomerization state cannot be determined. In view of the stable pentamer in the pORF19$_{KCTD}$ crystal structure, the oligomerization observed in AUC likely reflected a situation where protein concentration was too low to fully populate the pentameric state. These results indicated that the equilibrium between monomeric and pentameric pORF19$_{KCTD}$ is shifted toward the pentamer by the presence of high lithium acetate concentrations. Analysis of pORF19 oligomerization in a range of buffers and ionic strength conditions using sedimentation velocity experiments revealed that an oligomerization, albeit to a significantly lesser extent, can also be induced in the absence of lithium acetate (S5 Table), suggesting that the equilibrium can be shifted by various external triggers.

To engineer an oligomeric form that is stable under low salt conditions, we introduced an inter-protomeric disulfide bond by mutating 2 proline residues into cysteines (P$^{137}$C and P$^{461}$C; pORF19$_{KCTD}$CC, Fig 4B and 4C). Upon oxidation of cysteine thiols in recombinant pORF19$_{KCTD}$CC in 0.5 M lithium acetate, the oxidized protein formed a stable oligomer in solution (as shown by SEC; S2 Fig). The c(s) analysis of sedimentation velocity data in LiAc buffer revealed a single species sedimenting with 8.9 S (Fig 4D) and a molecular mass of 210 kDa, which is between that of a tetramer and a pentamer, and 2D classification of negative stain EM images of pORF19$_{KCTD}$CC unambiguously demonstrated the 5-fold symmetry of the the oxidized protein (Fig 4E and 4F, S6 Fig).

To better understand pORF19 pentamerization, we carried out structure-based mutagenesis of the pentameric lateral interface. We designed variants expected to have reduced inter-protomer binding based on 3 approaches: (I) removal of 2 prominent side chains contributing to the lateral interaction (D$^{173}$A/Q$^{177}$A; pORF19$_{KCTD}$DQ); (II) replacement of an interacting loop ($^{156}$MNQNQ$^{160}$) by a flexible linker (pORF19$_{KCTD}$loop); or (III) generation of steric clashes via insertion of bulky side chains (V$^{463}$W/L$^{465}$Y; pORF19$_{KCTD}$VL, Fig 4G–4I). The latter mutation was engineered into a surface loop that not only interacts with the adjacent protomer in the pentameric pORF19 ring, but was also proposed to constitute the main interface for the association of pORF19 with the penton pORF25 [4]. The respective mutant proteins were expressed, and sedimentation velocity experiments revealed that pORF19$_{KCTD}$DQ,

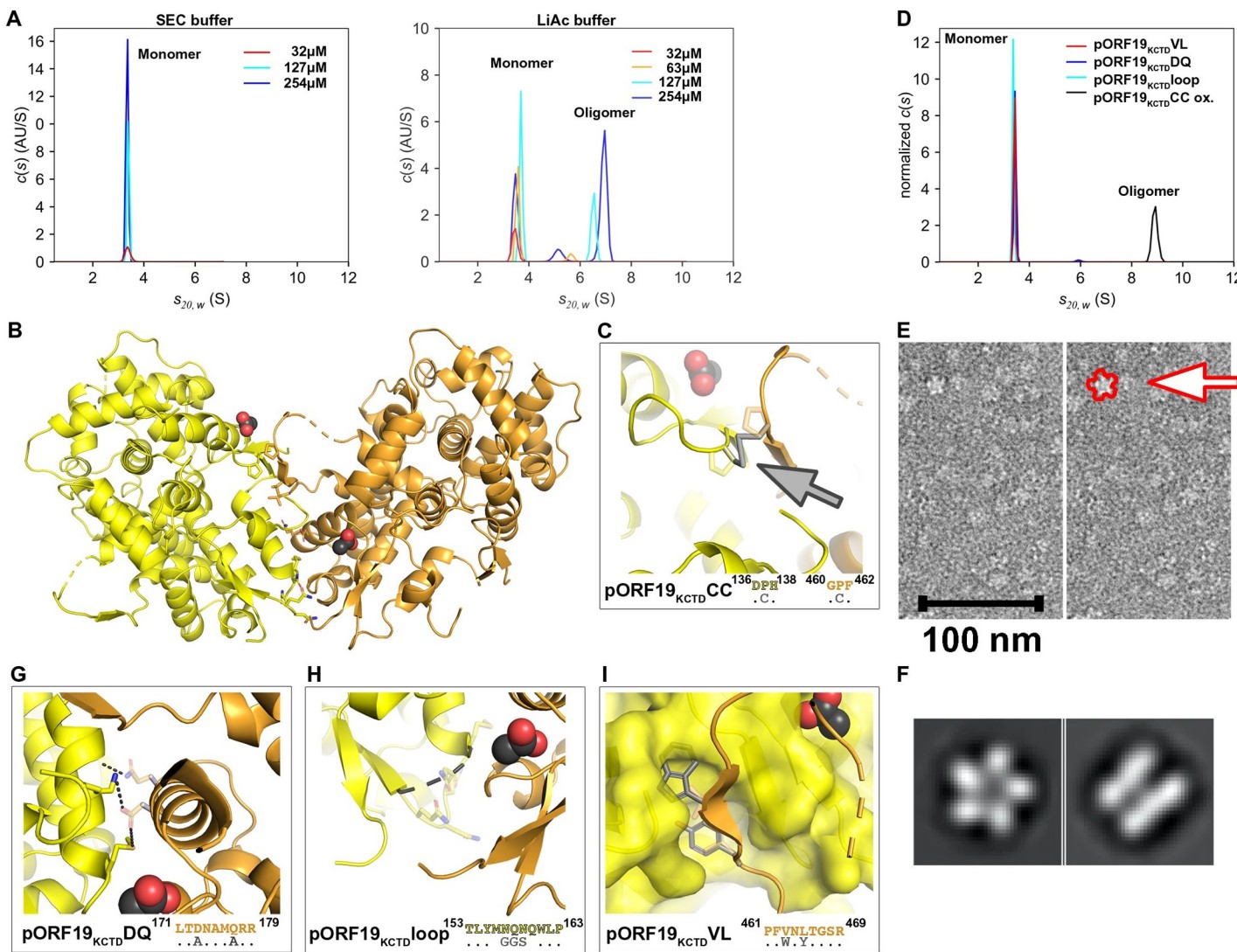

**Fig 4. Structure-based mutagenesis abrogates pORF19$_{KCTD}$ pentamerization in solution.** (A) Oligomerization of pORF19$_{KCTD}$ in SEC buffer (left panel; 10 mM Tris pH 7.5, 50 mM NaCl) and LiAc buffer (right panel; 100 mM HEPES pH 7.5 and 0.5 M lithium acetate). In SEC buffer, pORF19$_{KCTD}$ sediments with an s-value of 3.4 S independently of the used concentration. By contrast, in LiAc buffer the sedimentation coefficient of pORF19$_{KCTD}$ increases in a concentration dependent manner, indicating protein oligomerization. (B–I) Based on the 3D structure of the lateral pentameric interface (B), 4 mutants were designed to manipulate the pentamerization process. (C) To stabilize the pentamer a disulfide bond was engineered by introducing 2 cysteine residues according to the DisulfidebyDesign server [42]. (D) Oligomerization of pORF19$_{KCTD}$ mutants in LiAc buffer containing 50 mM HEPES. Under conditions where the wt protein shows significant, but not complete oligomerization (127 μM; panel A, right), the oxidized pORF19$_{KCTD}$CC (black curve) sedimented as a stable pentamer with 8.9 S even at a lower concentration of 11 μM. For better comparability of different protein concentrations, c(s) distributions were normalized to the same area. The underlying data for panels A and D can be found in S1 Data. (E) Negative stain EM of the oxidized pORF19$_{KCTD}$CC revealed a clear 5-fold symmetry, indicating that it is pentameric in SEC buffer. (F) Selected 2D classes representing a top and side view of oxidized pORF19$_{KCTD}$CC, suggesting that the latter has a high propensity to form a double layer "stacked ring" assembly in solution. (G–I) Three mutants with a destabilized lateral pentameric interface were generated based on (I) removal of prominent side chains contributing to the interaction (G, pORF19$_{KCTD}$DQ); (II) replacement of an entire interacting loop by a flexible linker (H, pORF19$_{KCTD}$loop); and (III) generation of steric clashes via insertion of bulky side chains (I, pORF19$_{KCTD}$VL). Interface mutants are shown in cartoon and/or surface representation with the observed crystal structure transparent and colored as in B and a computationally generated model of the introduced mutations in gray. Of note, all 3 interface mutants were virtually exclusively monomeric in AUC at a concentration of 127 μM, as indicated by an s-value of 3.4 S (see panel D), demonstrating that all mutations abolish oligomerization of the recombinant proteins. AUC, analytical ultracentrifugation; EM, electron microscopy; SEC, size exclusion chromatography; wt, wild-type.

pORF19$_{KCTD}$loop, and pORF19$_{KCTD}$VL were present virtually exclusively as monomers under conditions where the wild-type (wt) protein showed significant oligomerization (Fig 4A and 4D), indicating an efficient block of pentamerization.

## pORF19 is essential for the assembly of infectious KSHV progeny

Next, we investigated the role of pORF19 in the assembly of infectious KSHV virions using a knockout mutant lacking its globular domain and the pentamerization-incompetent variants described above. On KSHV capsids pORF19 can interact with vertices in 2 different ways— either (1) as heterodimer with pORF32 associating with the triplex around the penton vertex and directly interacting via its globular domain with pORF25; or (2) as homopentamer associating with the portal. We first analyzed whether mutant pORF19 proteins can interact with the penton vertices. For this purpose, this we employed a previously reported capsid assembly assay based on the production of KSHV capsids in insect cells [43]. In this assay, all essential KSHV capsid proteins (MCP pORF25, triplex proteins pORF62 and pORF26, protease pORF17, and scaffold protein pORF17.5 and SCP pORF65) except the portal protein pORF43 are produced from individual baculoviruses and contribute to the assembly of icosahedral KSHV capsids in insect cells. Upon co-expression of pORF19 and pORF32 (the second KSHV CVSC component), these assembled capsids lacking the portal vertex will incorporate both CVSC constituents [44], indicating an association of pORF19 with the penton vertices. In this setup, we co-expressed the essential KSHV capsid proteins together with pORF32 and either wt or mutant pORF19 proteins, respectively. In all cases, density gradient centrifugation of cell lysates revealed a clear capsid band at the expected position (see Materials and methods). Immunoblot analysis of gradient-purified capsids revealed that mutant pORF19 proteins were associated with capsids to a similar extent as wt pORF19 (Fig 5), suggesting that the pentamerization-blocking mutations did not affect the association of the CVSC with pentons under the conditions of this experiment.

To determine the biological relevance of pORF19 in the context of viral infection, we introduced a stop codon into a KSHV genome cloned in the bacterial artificial chromosome clone 16 (KSHV-Bac16) [45], resulting in a knockout of the carboxyl-terminal pORF19; we also introduced the pentamerization-blocking mutations described above into separate KSHV genomes. The sequence of the parental and the 4 mutant KSHV-Bac16 constructs (KSHV-Bac16DQ, KSHV-Bac16loop, KSHV-Bac16VL, and KSHV-Bac16 KO) was confirmed by next generation sequencing. We independently transfected iSLK cells with the parental

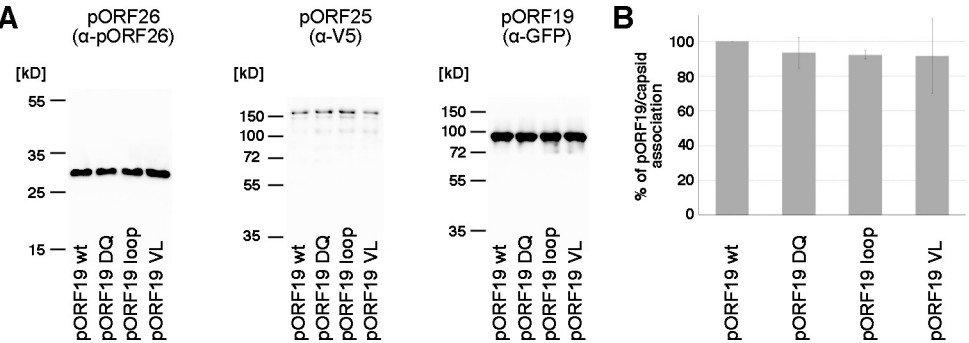

**Fig 5. Association of mutant pORF19 proteins with KSHV capsids produced in insect cells. (A)** Baculovirus-produced KSHV capsids lacking the portal protein pORF43 and containing either wt or mutant pORF19 were purified by gradient ultracentrifugation from cell lysates as described before [44] and analyzed by western blot. The quantity of capsids loaded was normalized to the MCP pORF25 carrying a V5 tag (153.4 kD) and the triplex protein pORF26 (30 kD). Immunoblot analysis of GFP-tagged pORF19 of a representative experiment is shown (88.2 kD). **(B)** Capsid productions were performed in 3 independent biological replicates, intensities of pORF19 bands were quantified, expressed in percent of wt pORF19, and mean values with error bars representing SD shown. The underlying data can be found in S2 Data. KSHV, Kaposi's sarcoma-associated herpesvirus; MCP, major capsid protein; SD, standard deviation; wt, wild-type.

KSHV-Bac16 and the 4 mutant constructs. These cells are derived from SLK endothelial cells and allow for efficient virus reactivation and production of substantial quantities of infectious KSHV upon induction with doxycycline, resulting in the expression of the lytic switch protein RTA [46]. Stable cell clones were selected using the hygromycin resistance gene provided by KSHV-Bac16 and individual cell clones, in which KSHV reactivated upon induction were identified by immunoblot using antibodies directed against the early lytic protein K-bZIP (Fig 6A). To analyze the production of infectious progeny, we collected supernatants after reactivation. Titration of these supernatants on HEK293 cells revealed $1.4 \times 10^5$ infectious units per ml for the parental KSHV-Bac16 construct, while iSLK cells transfected with the KSHV-Bac16 KO (expressing a pORF19 protein lacking the globular domain) did not produce any infectious progeny above background level as defined by supernatants from non-transfected iSLK cells used as control (Fig 6B). This is similar to a previously reported pUL25 knockout mutant in HSV-1 that was generated by insertion of a stop codon in the UL25 reading frame at a similar position upstream of its globular domain [22].

Mutants KSHV-Bac16VL (expressing pORF19 with steric clashes in the lateral interface) and KSHV-Bac16loop (expressing pORF19 lacking an interacting loop) did not produce any infectious progeny—like the knockout mutant—while supernatants from cells transfected with mutant KSHV-Bac16DQ contained infectious progeny, albeit approximately 140× less than the parental KSHV-Bac16 construct. It is tempting to speculate that the more pronounced inhibition observed for KSHV-Bac16loop and KSHV-Bac16VL when compared to KSHV-Bac16DQ could result from secondary effects. For example, the mutation in pORF19$_{KCTD}$VL could not only block pentamerization, but also the interaction with the penton pORF25 proposed previously [4]. Further functional and structural analysis will be required to analyze the mechanisms of inhibition in more detail.

To confirm that the observed effect of the pentamerization-blocking mutations was not due to inadvertent mutations in our KSHV-Bac16 mutants, we *trans*-complemented the stable KSHV-Bac16 mutant transfected cell clones with wt pORF19. For this purpose, we transduced individual cell clones with retroviral particles encoding both pORF19 wt and a red fluorescent protein (RFP) under the control of a minimal CMV promoter [47,48] and sorted them for homogeneous RFP expression by fluorescence-activated cell sorting (FACS). The resulting cell populations produced approximately 8 to 30× more infectious particles for each individual KSHV-Bac16 mutant construct upon *trans*-complementation of wt pORF19 (Fig 6B). By contrast, *trans*-complementation with RFP alone did not increase the production of infectious KSHV progeny. Of note, none of the complemented cell populations reached titers similar to iSLK cells transfected with wt KSHV-Bac16. Such an incomplete *trans*-complementation by wt pORF19 can be expected, if the mutant pORF19 protein that is also expressed in these cells acts as dominant-negative competitor in pORF19 pentamerization.

The lack of infectious particles in the supernatant of iSLK cells transfected with mutant KSHV-Bac16 can be either due to an impaired viral particle formation (i.e., a block in capsid assembly) or a lack of infectivity (i.e., a block in cell entry and nuclear targeting of incoming capsids). To distinguish between these possibilities, we purified KSHV particles from supernatants of individual iSLK cell clones using density gradient centrifugation. Particles released from cells transfected with wt KSHV-Bac16 and KSHV-Bac16DQ contained the viral envelope (gH), viral tegument (pORF45), and the viral capsid (pORF26) as revealed by immunoblot. By contrast, the corresponding gradient fractions from cells transfected with KSHV-Bac16VL, KSHV-Bac16loop, and KSHV-Bac16 KO used as control did not contain gH, pORF45, or pORF26, in line with their more strongly impaired production of infectious progeny. Of note, upon complementation of these cell clones with wt pORF19, the marker proteins gH, pORF45, and pORF26 were readily detected in the corresponding gradient fractions (Fig 6C). These

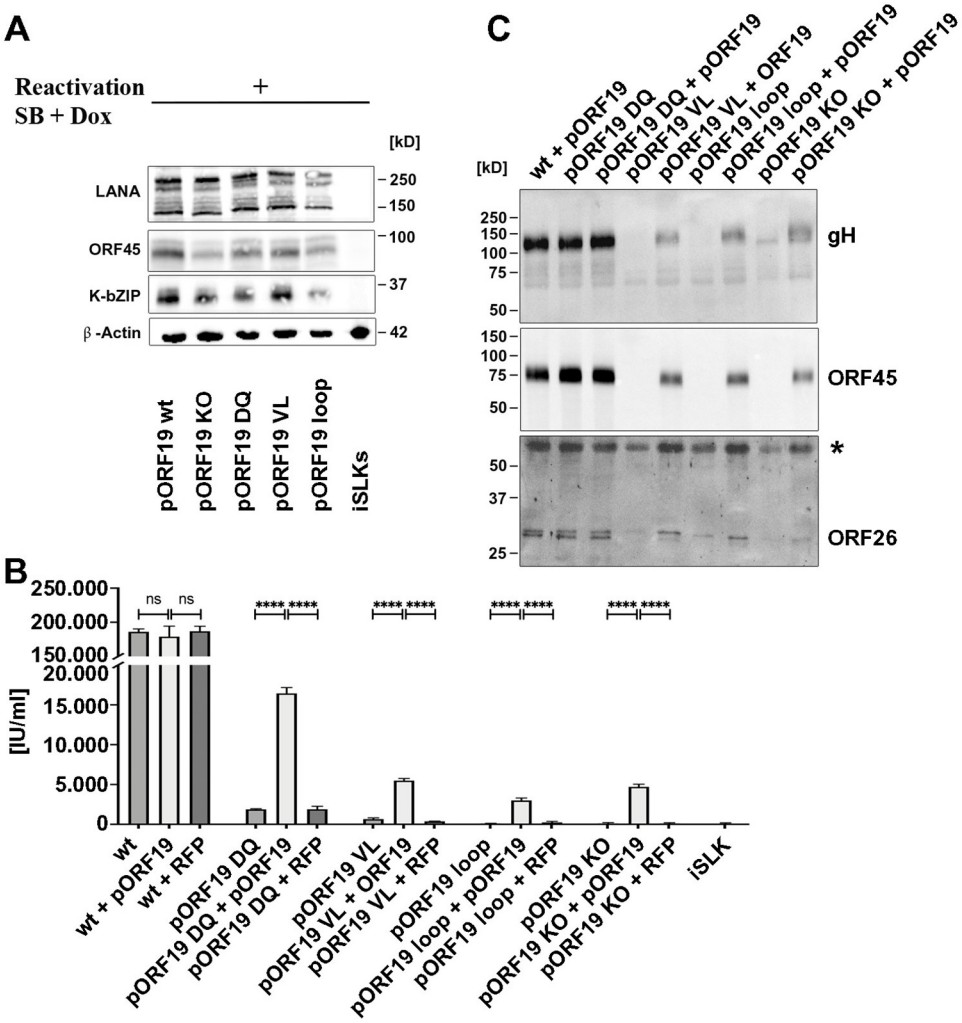

**Fig 6. pORF19 pentamerization is important for production of infectious progeny. (A)** KSHV reactivation in individual iSLK cell clones transfected with either wt or mutant KSHV-Bac16 constructs was assessed by immunoblot analysis of the early lytic protein K-bZIP (37 kD). **(B, C)** iSLK cell clones carrying either wt or mutant KSHV-Bac16 constructs were transduced with retroviral particles encoding either pORF19 and RFP (wt + pORF19), TRIM5α and RFP (wt + RFP) used as control or not transduced (wt) to analyze functional pORF19 complementation. Transduced cells were sorted by FACS using the RFP to obtain cell populations displaying homogeneous protein expression, and KSHV was reactivated in the resulting populations. **(B)** Titration of the infectious particles present in the respective supernatants revealed an 8 to 30× increase in the production of infectious progeny upon functional complementation for all 3 mutants. Titrations were performed in triplicate of 3 biological replicates, and the mean ± SD was calculated. The underlying data can be found in S2 Data. Statistical comparisons were performed via 1-way ANOVA with Bonferroni correction. ****$P < 0.0001$. n.s., not significant. **(C)** Immunoblot analysis of gradient-purified KSHV particles from the supernatants of individual cell clones and their complemented counterparts using antibodies specific for glycoprotein gH (viral envelope; 130 kD), pORF45 (viral tegument; 78 kD) and the triplex protein pORF26 (viral capsid; 30 kD) confirmed the more pronounced defect in virus particle release observed for iSLK cells transfected with KSHV-Bac16VL or KSHV-Bac16loop when compared to KSHV-Bac16DQ. The asterisk indicates an unspecific band that was always observed in iSLK cells. FACS, fluorescence-activated cell sorting; KSHV, Kaposi's sarcoma-associated herpesvirus; RFP, red fluorescent protein; wt, wild-type.

results suggest that cells containing KSHV-Bac16VL-, KSHV-Bac16loop-, or KSHV-Bac16 KO- do not support the production of virions and that a functional pORF19 is essential for the release of KSHV particles.

### pORF19 mutants impact either on capsid assembly or DNA packaging/ retention in KSHV-infected cells

Next, we determined the subcellular localization of the capsid triplex 2 protein pORF26 in the different KSHV-Bac16 cell lines by immunofluorescence microscopy (Fig 7A). In cells transfected with the wt KSHV-Bac16 (Fig 7A top line), and to a lesser extent in cells with KSHV-Bac16DQ (Fig 7A second line), the capsid triplex protein pORF26 was predominantly localized in confined regions of the nucleoplasm that resemble nuclear capsid assembly sites as described before [49,50]. Furthermore, these cells contained numerous individual pORF26 puncta in the cytoplasm, which might reflect KSHV capsids after nuclear egress. By contrast, no pORF26 signal was observed in the cells transfected with KSHV-Bac16VL, KSHV-Bac16loop, or KSHV-Bac16KO (Fig 7A). However, *trans*-complementation with wt pORF19

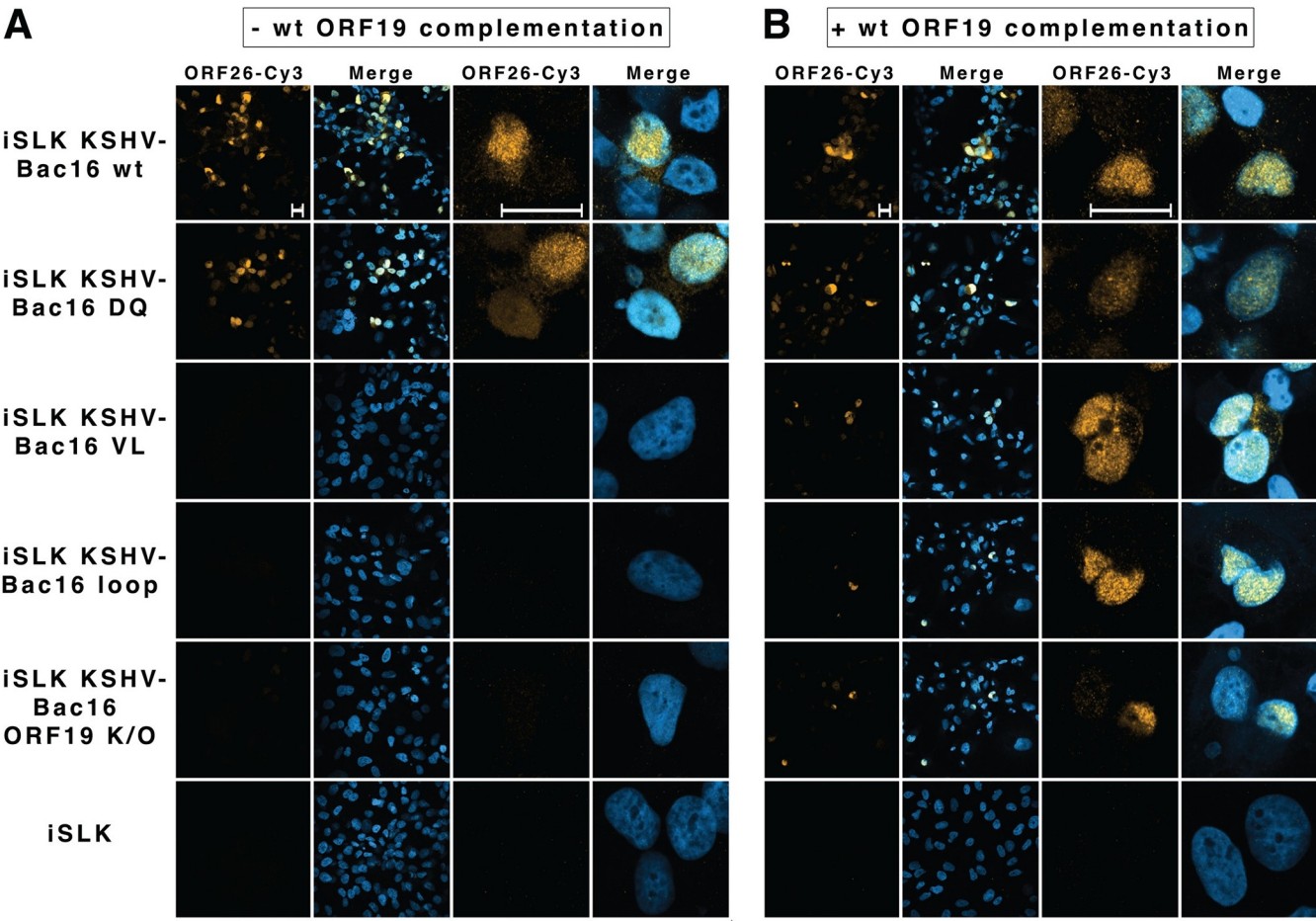

**Fig 7. Functional pORF19 is required for capsid assembly.** Individual iSLK cell clones transfected with and without KSHV-Bac16 constructs prior to **(A)** and after complementation with wt pORF19 **(B)** were analyzed after KSHV reactivation by immunofluorescence microscopy using an antibody targeting the triplex protein pORF26 (orange) and DAPI (blue) to stain nuclei. The left 2 columns of each panel depict overview pictures, and the right 2 columns of each panel depict higher magnification pictures focusing on individual cells. **(A)** In wt KSHV-Bac16 and to a lesser extent KSHV-Bac16DQ transfected cells a typical intranuclear distribution of the triplex protein pORF26 was observed, corresponding to the accumulation of capsids in confined regions of the nucleoplasm that resemble nuclear capsid assembly sites. The presence of pORF26 puncta in the cytoplasm indicates nuclear egress of capsids. By contrast, no nuclear pORF26 signal was detected in cells transfected with the other mutants or the KSHV-Bac16 pORF19 KO construct. **(B)** Complementation with wt pORF19 using retroviral transduction restored the intranuclear pORF26 staining, thereby demonstrating that pORF26 accumulation within the KSHV nuclear capsid assembly sites depends on the presence of functional pORF19. Nuclear egress is also observed upon complementation in these cells, albeit to a lower extent than for wt KSHV-Bac16 transfected cells. Shown images are stacked images and individual panels were generated with ImageJ. Scale bars correspond to 20 µm. KSHV, Kaposi's sarcoma-associated herpesvirus; wt, wild-type.

restored a prominent nuclear labeling for pORF26 also in cells with KSHV-Bac16VL, KSHV-Bac16loop, or KSHV-Bac16KO (Fig 7B), and few cytoplasmic pORF26 puncta were present in these cells (Fig 7B). These results suggested that functional pORF19 is essential for the formation of nuclear and cytoplasmic puncta, most likely capsids that contained pORF26 in a mature conformation recognized by the used antibody. Formally the observed lack of pORF26 signal could also be due to mutated pORF19 proteins hampering pORF26 expression, inducing pORF26 degradation, or impairing the formation of icosahedral capsids. However, angular KSHV capsids assemble in the context of a capsid assembly assay using baculovirus capsid protein overexpression in spite of the absence of pORF19 [43].

We therefore investigated KSHV morphogenesis and capsid assembly by electron microscopy. In cells transfected with wt KSHV-Bac16 (Fig 8Ai), the nucleus contained numerous empty A capsids (Fig 8Aii), capsids containing a protein-core (B capsids; Fig 8Aiii), and C capsids containing viral genome (Fig 8Aiv) and predominantly C capsids underwent primary envelopment at the inner nuclear membrane (white arrows in Fig 8A). Nuclei of cells transfected with KSHV-Bac16DQ contained A- and B-capsids, but no mature C capsids or capsids undergoing primary envelopment were observed, although cells transfected with this KSHV mutant produced infectious particles to some extent (Fig 6B). By contrast, we did not detect any capsids in cells transfected with KSHV-Bac16VL (not shown), KSHV-Bac16loop (Fig 8C and 8D), or KSHV-Bac16 KO (not shown), in line with the lack of infectious particles being secreted from the respective reactivated cell clones (Fig 6B).

In insect cells, baculovirus infection results in massive target protein overexpression of up to 50% of total cellular protein [51], in case of herpesviral capsid proteins leading to capsid assembly also in the absence of the portal protein pORF43 [52]. It is likely that in infected cells the expression level does not reach equivalent levels, since a number of viral proteins together with a plethora of cellular proteins involved in essential cellular processes are expressed simultaneously. These considerations indicate that a capsid assembly assay upon baculovirus infection in insect cells reflects capsid assembly in infected cells only to a limited extent. Our results described above indicate that under physiologic conditions an intact pORF19 is key for capsid assembly (Fig 8) and for the production of infectious KSHV virions (Fig 6B). As the KSHV-Bac16DQ mutant, which interferes with pORF19 pentamerization (Fig 4), still allowed the assembly of empty A and B capsids but not DNA-containing C capsids (Figs 6–8), pentamerization of pORF19 at the portal cap may be required for correct DNA packaging. These insights draw attention to the lateral pentameric pORF19 interface as a promising target for the development of novel antiviral inhibitors.

## Discussion

Recent symmetry-relaxed cryo-EM reconstruction strategies of α- and γ-herpesviral virions provided first snapshots of the unique portal vertex that is essential for DNA encapsidation and release [4–6]. These studies revealed for the first time a pentameric cap on top of the portal that presumably seals the capsids against DNA leakage after packaging and has been proposed to consist of pUL25 in HSV-1 and its ortholog pORF19 in KSHV [4–6]. However, the structure of the portal cap has remained elusive to date, as the lower resolution of the cryo-EM electron density maps in this region did not allow to build an atomic model. Our crystal structure of the pentameric globular domain of KSHV pORF19 at 2.6 Å resolution could now close this gap, as the rigid body fitting of the pentamer into the cryo-EM map of the KSHV portal vertex [4] strongly suggest a physiological relevance of our pentameric pORF19$_{KCTD}$ structure (Fig 3). Cryo-EM reconstructions of HSV-1 and KSHV indicate an anchoring of the pUL25/pORF19 cap to the capsid via the so-called "tentacle helices" emanating from the portal clip, in

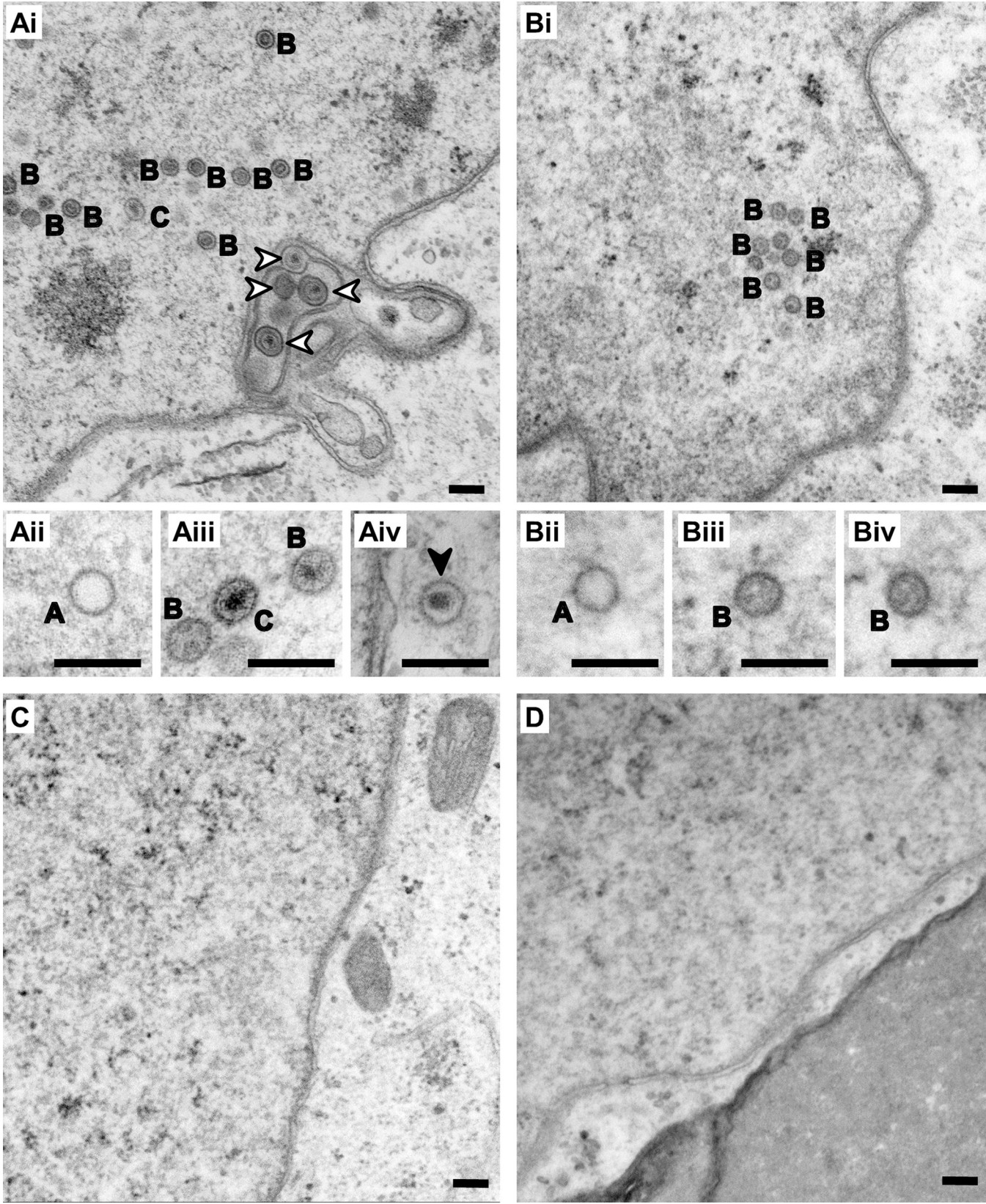

**Fig 8. Formation of capsids in reactivated cells requires the presence of functional pORF19. (A–D)** Electron microscopy analysis of individual iSLK cell clones transfected with either wt or mutant KSHV-Bac16 constructs after KSHV reactivation. In cells transfected with wt KSHV-Bac16 **(A)** viral capsids were frequently found in the nucleus (Ai, Aii, and Aiii) and to a lesser extent in the cytoplasm (Aiv). In the nucleus, all stages of nuclear capsid assembly were identified, including A capsids (labeled with A), B capsids (labeled with B), and C capsids (labeled with C), as well as capsids in the process of primary envelopment at the nuclear envelope (Ai, white arrowheads). In cells transfected with KSHV-Bac16DQ **(B)**, only nuclear capsids of the types A (labeled A in Bii) and B (labeled B in Biii and Biv), but no C capsids were observed. No capsids were found undergoing primary envelopment or in the cytoplasm. In cells transfected with KSHV-Bac16VL (C) or KSHV-Bac16loop (D), no viral structures were identified in the nucleus or the cytoplasm, similar to mock-transfected cells. Scale bars correspond to 200 nm. KSHV, Kaposi's sarcoma-associated herpesvirus; wt, wild-type.

line with the reported direct interaction of pUL25 with the portal protein pUL6 in HSV-1 [53]. The remarkable asymmetric charge distribution on one face of the pORF19 pentamer explains the apparent interaction between the last 5 terminal base pairs of the genome and the portal cap revealed in a recent cryo-EM reconstruction of HSV-1 [5]. It is tempting to speculate that strong electrostatic interactions between the positively charged pORF19 funnel and the negatively charged DNA backbone tightly seal the portal channel after packaging, making these interactions therefore essential for stable genome encapsidation. Of note, only one lysine residue creating these positive charges in the funnel is conserved across herpesviruses, suggesting that other positively charged residues generate a similarly charged surface. It will therefore be interesting to investigate whether similar electrostatic interactions can be observed in the pentameric forms of pORF19 orthologs.

Of note, the presence of pORF19 is essential for capsid formation as demonstrated by the absence of assembled capsids in iSLK cells transfected with a pORF19 knockout mutant (see above). This is in stark contrast to results obtained for HSV-1 with a previously reported pUL25 knockout mutant that allowed assembly of A- and B- capsids [22], therefore illustrating functional differences between KSHV pORF19 and HSV-1 pUL25. Our results indicate that pORF19 is required for the formation of icosahedral capsids; however, it remains elusive, to date, whether pORF19 is directly involved in the formation of icosahedral capsids prior to stabilization by the CVSC, in the recruitment of other capsid proteins to the assembly site, in stabilization of other capsid proteins, or whether pORF19 acts via an unanticipated indirect effect. Careful functional studies on pORF19 are required to decipher the precise role of pORF19 during KSHV assembly. Such studies are likely to better illustrate the functional differences between γ-herpesvirus pORF19 and α-herpesvirus pUL25 described here.

The striking structural conservation between pORF19, pUL25, and their β-herpesviral ortholog pUL77 most likely indicates an evolutionary conserved function that depends on this conserved structure. pUL77 associates with HCMV capsids residing in the nucleus and is also found in extracellular virions [28], but unlike in α- and γ-herpesviruses, symmetrical cryo-EM reconstructions of β-herpesviruses do not reveal any density that could correspond to the pUL77 globular domain [18,19]. The comparison across these 3 herpesviral subfamilies indicates that rather the pentameric assembly capping the portal than the heterodimeric CVSC associated with the penton vertex provides the key conserved function to stably package the viral DNA genomes into the capsids. Our results therefore also suggest the presence of a pentameric pUL77 assembly capping the HCMV portal, in line with the reported tendency of pUL77 to oligomerize [54].

One intriguing question remains: What triggers the oligomerization of nuclear KSHV pORF19, HSV-1 pUL25, and, possibly, HCMV pUL77 to seal DNA-filled C capsids, and how are the pentameric caps removed at the nuclear pores to allow release of incoming genomes into the nucleus for the next infection? The CVSC proteins along with the terminase components are believed to assemble onto the capsid before the start of DNA encapsidation (reviewed in [16]). Recent studies on HSV-1 capsids suggest a coordinated multistep process involving an ordered interaction of the terminase complex with pUL17, which, together with

pUL25, forms the CVSC, and the pUL6 portal that leads to the cleavage of the concatameric genomes and packaging of one genome per capsid. The subsequent release of the terminase goes along with an outward displacement of the portal by 30 Å [31] to facilitate sealing of the portal by a pentameric pUL25. One hypothesis could therefore be that pORF19 pentamerizes (1) spontaneously; or (2) triggered by terminal DNA around the portal vertex in a process that is initiated by its CTD once the terminase complex has dissociated, in line with the pentamerization of our pORF19$_{KCTD}$. Of note, such a pentamerization would be sterically more difficult around the penton vertex, since the latter protrudes further from the capsid floor than the portal and pentamerization via the CTD would therefore require major conformational rearrangements in the CVSC helical bundle and/or triplex association.

Pentameric portal caps have been demonstrated in α- and γ-herpesviruses [4,6,18]; however, neither HSV-1 pUL25$_{CTD}$ [17] nor MuHV-68 pORF19$_{MCTD}$ are pentameric in solution or in the crystal, suggesting that additional triggers or stabilizing forces are required for pentamerization. One possible factor contributing to pentamer stability in vivo could be the N-terminal segment, which is absent in the crystallized CTDs of all orthologs. However, on capsids, these segments are part of the CVSC helical bundles symmetrically distributed around the portal vertex, thus rendering a direct contact that could stabilize the pentamer unlikely [4–6,11,12,15,20,55]. On the other hand, a high local concentration of pORF19 CTDs determined by the 5 symmetric CVSC copies that can directly interact at the portal—in contrast to the more extensively protruding penton vertex—likely contributes to pentamer stabilization, in line with the concentration dependent pentamerization of pORF19$_{KCTD}$ in vitro. In addition, our pORF19$_{KCTD}$ pentamer structure reveals the presence of 2 acetate ions in the lateral interface that stabilize the pentamer both in the crystals and in solution. In the nucleus, acetate ions are continuously snatched or released from histones in a highly regulated balance between the activities of histone acetyltransferases (HATs) and histone deacetylases (HDACs), serving as a key regulatory mechanism for gene expression [56]. We can therefore not formally exclude that acetate ions are also required for KSHV portal cap formation in vivo. However, the high acetate ion concentrations required for pentamerization in solution together with the fact that buffers lacking acetate ions also induced pORF19$_{KCTD}$ oligomerization and both MgAc$_2$ and NaAc buffers induced oligomerization to a much lower extent (S5 Table) suggest that different triggers can shift the monomer-oligomer equilibrium at least in vitro. The identity of the trigger that determines pentamerization in vivo remains elusive to date.

The portal cap allows the capsid to withstand the high capsid pressure caused by repulsive forces of the encapsidated viral DNA genome [25,57]. This indicates that (1) a tight interaction between portal and portal cap is required to prevent premature DNA leakage; and (2) that disruption of this interaction, i.e., disassembly or removal of the portal cap, is a highly coordinated process that leads to the release of the viral genome through the nuclear pore complexes into the nucleoplasm [58,59].

Lastly, our results demonstrate that the lateral pORF19 interface constitutes an attractive target for future development of herpesvirus inhibitors. Currently, most available treatment options against herpesvirus-induced diseases target the viral polymerase and are often associated with severe side effects (reviewed in [2]). The unique character of the portal vertex and the possibility to block its sealing by targeting the lateral interface of the KSHV pORF19 pentamer (Fig 6) opens up an exciting perspective of generating potent novel antivirals using structure-based drug design. While the benefit of such an inhibitor of pORF19 pentamerization in therapy of KSHV-induced diseases remains to be determined, the portal cap is conserved in α-herpesviruses and likely also in β-herpesviruses, where the development of additional therapeutic options is an urgent medical need.

## Materials and methods

### Construct design and cloning

Expression constructs of pUL77$_{CTD}$, pORF19$_{KCTD}$, and pORF19$_{MCTD}$ comprising residues 180 to 642 (HCMV pUL77), 123 to 549 (KSHV pORF19), and 104 to 516 (MuHV-68 pORF19), respectively, were designed in analogy to the published crystal structure of a carboxyl-terminal fragment of HSV1 pUL25 [17] based on an amino acid alignment using the PROMALS 3D server [60]. The gene fragments were cloned into a pET28 based vector either using *NcoI* and *NotI* restriction sites (for pUL77$_{CTD}$ and pORF19$_{MCTD}$) or by restriction-free cloning (for pORF19$_{KCTD}$) [61]. pORF19$_{KCTD}$ mutants were generated by Quik-Change site directed mutagenesis. All primers used throughout this studies are listed in S6 Table.

### Protein expression and purification

pUL77$_{CTD}$, pORF19$_{KCTD}$, pORF19$_{MCTD}$, and pORF19$_{KCTD}$ mutants were expressed in the cytoplasm of transformed *Escherichia coli* Rosetta (DE3) cells for 20 hours at 16˚C after induction with IPTG (0.2 mM) at an OD$_{600}$ of 0.6. For expression of a selenomethione derivative of pORF19$_{KCTD}$, an overnight pre-culture derived from an individual transformed *E. coli* Rosetta (DE3) colony was pelleted, washed twice with M9 minimal medium, and resuspended in 1-L fresh M9 medium to inoculate the expression culture at 37˚C until an OD600 of 0.5 was reached. Subsequently, an amino acid cocktail (100-mg phenylalanine, 100-mg threonine, 50-mg isoleucine, 50-mg leucine, 50-mg valine, and 100 mg lysine) was added, and, after an incubation of 30 minutes at 30˚C, the cultures were induced with 0.2 mM IPTG, and 60 mg of selenomethionine was added. The selenomethionine derivative was expressed for 24 hours at 16˚C.

For protein purification, pelleted bacteria were resuspended in extraction buffer (50 mM Tris pH 7.5, 300 mM NaCl, 40 mM imidazole, 10% glycerol) and lysed using a Constant Systems cell disruptor operating at approximately 21.8 kpsi. The clarified lysate was loaded onto a 1 ml HisTrap crude column (GE Healthcare, Solingen, Germany), which was pre-equilibrated in the respective extraction buffer. Purified pUL77$_{CTD}$, pORF19$_{KCTD}$, and pORF19$_{MCTD}$ was eluted in 500 mM imidazole followed by SEC on a Superdex 200 Increase 10/300 GL column (GE Healthcare) in SEC buffer (10 mM Tris pH 7.5) containing high salt (1 M NaCl; pUL77$_{CTD}$) or low salt (50 mM NaCl; pORF19$_{KCTD}$ and pORF19$_{MCTD}$). The column was calibrated using an molecular weight (MW) standard and a calibration curve was generated from the corresponding elution volumes to allow for an approximation of the oligomeric state of the recombinant proteins.

Purified proteins after initial SEC were concentrated to 1 mg/ml and digested with carboxypeptidase (Sigma, Taufkirchen, Germany) to remove the carboxyl-terminal Histidine tag according to the manufacturer's recommendations. Digested protein was separated from the undigested protein by filtration through a 0.45 μM syringe filter followed by a 2-step purification procedure. First, the protein was loaded onto a 1-ml HisTrap crude column, and the flow-through was further purified by SEC on a Superdex 200 Increase 10/300 GL column (GE Healthcare) in SEC buffer (10 mM Tris pH 7.5) containing high salt (1 M NaCl; pUL77CTD) or low salt (50 mM NaCl; pORF19KCTD and pORF19MCTD).

### Crystallization and structure determination

Crystals for all proteins were grown at 18˚C. pUL77$_{CTD}$ crystals were grown using the sitting drop vapor diffusion method in a drop containing 0.25 μl of pUL77$_{CTD}$ (6 mg/ml) and 0.25 μl of reservoir containing 0.08 M Na-Cacodylate pH 6.5, 16% PEG 8000, and 20% glycerol,

belonged to the tetragonal $P4_122$ spacegroup and diffracted to 1.9 Å. The structure was determined by the molecular replacement method using PHASER [62] and a search model derived from the structure of pUL25 (PDB 2F5U, [17]).

Native pORF19$_{KCTD}$ crystals were grown using the hanging drop vapor diffusion method in a drop containing 1 μl of pORF19$_{KCTD}$ (6 mg/ml) and 1 μl of a reservoir solution containing 0.1 M HEPES pH 7.5 and 1.69 M lithium acetate. pORF19$_{KCTD}$ crystallized in the $P2_12_12_1$ spacegroup and the crystals diffracted to 2.6 Å. In contrast to the crystals of pUL77$_{CTD}$ and pORF19$_{MCTD}$, which contained one molecule in the asymmetric unit, pORF19$_{KCTD}$ crystals contained approximately 10 molecules per asymmetric unit, rendering structure determination by the molecular replacement method difficult. We therefore grew selenomethionine pORF19$_{KCTD}$ derivative crystals by the sitting drop vapor diffusion method in a drop containing 0.5 μl of the selenomethionine pORF19$_{KCTD}$ (6 mg/ml) and 0.5 μl of the reservoir solution containing 0.1 M MES pH 7.4 and 2 M lithium acetate. We obtained derivative crystals in 2 different spacegroups ($P2_1$ and $P6_322$) diffracting to 3.3 Å and 3.2 Å, respectively, and we determined the crystal structure by the multi-wavelength anomalous diffraction method using autoSHARP [63]. The initial electron density map revealed 1 and 2 pentameric pORF19$_{KCTD}$ assemblies per asymmetric unit, respectively. Molecular replacement using PHASER [62] and such a pentameric assembly as search model allowed refinement of the structure against the native dataset at 2.6 Å.

Crystals of pORF19$_{MCTD}$ were grown using the sitting drop vapor diffusion method in a drop containing 1 μl pORF19$_{MCTD}$ (6 mg/ml) and 1 μl of reservoir solution containing 2% PEG 6000 and 0.1 M HEPES pH 7.5 and belonged to the hexagonal $P6_1$ spacegroup. The best crystals diffracted to 1.9 Å, and the structure was determined by the molecular replacement method using PHASER [62] and a search model derived from the pORF19$_{KCTD}$ structure.

Data collection for all crystals was carried out at the Synchrotron Soleil (Proxima1 and Proxima2). Data were processed, scaled, and reduced with XDS [64], Pointless [65], and programs from the CCP4 suite [66]. Model building was performed using Coot [67], and refinement was done using AutoBuster [68] with repeated validation using MolProbity [69].

## Generation of pentameric pORF19$_{KCTD}$CC

To engineer a covalently linked pentameric pORF19$_{KCTD}$, we mutated 2 proline residues into cysteines as suggested by the DisulfidebyDesign server [42]. The corresponding recombinant protein (pORF19$_{KCTD}$CC) was expressed and purified as described above for pORF19$_{KCTD}$. Oxidized pORF19$_{KCTD}$CC was generated by reconstitution of pORF19$_{KCTD}$CC to a final concentration of 3 mg/ml in an oxidation buffer (0.1 M HEPES pH 7.5, 0.5 M lithium acetate, 2 mM oxidized Glutathion (GSSG)) and incubated at 37˚C for 16 hours. The oxidized protein was further purified by SEC using a Superose 6 Increase 10/300 GL column.

## Analytical ultracentrifugation

Runs were carried out in an analytical ultracentrifuge ProteomeLab XL-I (Beckman Coulter, Krefeld, Germany) using an An-50 Ti rotor at 40,000 rpm and 20˚C. Concentration profiles were measured with the manufacturer's data acquisition software ProteomeLab XL-I Version 6.0 (Firmware 5.7) using the absorption scanning optics at 280 nm or 290 nm. Standard 3-mm or 12-mm double sector centerpieces were filled with 100- or 400-μl sample, respectively. In the presence of lithium acetate, titanium centerpieces (Nanolytics Instruments, Potsdam, Germany) were used, otherwise charcoal-filled Epon centerpieces (Beckman Coulter). Experiments were performed in 10 mM Tris pH 7.5 and 50 mM NaCl, 100 mM HEPES pH 7.5 and

0.5 M lithium acetate, or 50 mM HEPES pH 7.5 and 0.5 M lithium acetate as indicated or in buffers containing different salt types and concentrations as indicated in S5 Table.

For data analysis, a model for diffusion-deconvoluted differential sedimentation coefficient distributions (continuous c(s) distributions) implemented in the program SEDFIT [70] was used. Partial specific volumes, extinction coefficients at 280 nm, buffer densities, and buffer viscosities were calculated from amino acid and buffer composition, respectively, by the program SEDNTERP [71] and were used to correct experimental s-values to $s_{20,w}$. Density ($\rho$ = $1.023 \cdot 10^3$ kg/m$^3$) and viscosity ($\eta$ = 1.262 mPa·s) of the buffer containing 50 mM HEPES pH 7.5 and 0.5 M lithium acetate at 20°C were measured using an areometer (Ludwig Schneider) and a Schott KPG Ubbelohde capillary viscosimeter with automatic sampler [72]. For the buffer containing 100 mM HEPES pH 7.5 and 0.5 M lithium acetate, an $s_{20,w}$ correction factor of 1.449 was determined by comparison of the experimental s-value of the monomer at 32 µM pORF19$_{KCTD}$ and $s_{20, w}$ of the monomer in the same buffer but with 50 mM HEPES. Protein concentrations were determined spectrophotometrically and are given for the monomers throughout the text. Figures were prepared using the program GUSSI [73].

### Structure analysis

Buried solvent accessible surface areas for the interfaces within the pentameric pORF19$_{KCTD}$ were calculated using jsPISA [74]. Interactions were determined using PIC [75]. Visualization of herpesviral capsid cryo-EM maps, rigid body fitting of the crystal structures into these maps were performed with the "SegFit" tool of UCSF Chimera [40]. During the fitting, the option for generation of a simulated map from the crystal structure at 7.6 Å was used. A cross-correlation coefficient between the simulated map and the experimental map was calculated and reported as a quantitative estimate of the quality of fitting. Figures were prepared with Pymol (http://www.pymol.org) and UCSF Chimera [40].

### Capsid assembly assay

pORF19$_{KCTD}$ mutants were generated using the pFastBac1 (pFB1) baculovirus transfer vector containing ORF19-GFP [44]; recombinant bacmids were produced following standard protocols and transfected into Sf9 cells using Effectene (Qiagen, Hilden, Germany) and a protocol developed for the transfection of *Drosophila* S2 cells [76], and 2 rounds of virus amplification were performed in Sf9 cells. Expression of individual proteins was confirmed 72 hours after infection by either immunoblot or SDS-PAGE followed by Coomassie staining. For capsid assembly experiments, Sf9 cells were coinfected with a mixture of all 7 distinct baculoviruses (encoding pORF25, pORF17, pORF17.5, pORF26, pORF62, pORF65, and pORF32)) together with the one encoding either wt or mutant pORF19. Cells were lysed in capsid extraction buffer (CEB: 500mM sodium chloride, 1mM EDTA, 20mM Tris-Hcl pH 7.5, 1% TritonX-100, Protease inhibitors (Roche, Mannheim, Germany) 1 tablet per 50 ml), further disrupted in a precooled douncer, and sonicated using a sonopuls 3200 set to 60% 10 times for 5 seconds with a 30-second pause on wet ice. The lysate was then centrifuged at 75,000 g for 90 minutes, the pellet resuspended in 5 ml CEB, gently layered onto a 20% to 50% sucrose continuous gradient (15 ml of 20% sucrose + 15 ml of 50% sucrose) and centrifuged at 50,000 g for 90 minutes. After centrifugation, a light scattering band at approximately 15 ml from the bottom of the sucrose gradient corresponding to the capsids was harvested as described before [44] for further analysis.

### Antibodies

The primary and secondary antibodies used for western blot analysis are listed below. The mouse anti-KSHV K-bZIP (sc-69797) was purchased from Santa Cruz Biotechnology

(Heidelberg, Germany). The primary mouse antibody anti-β-actin (A2228) was purchased from Sigma-Aldrich (Taufkirchen, Germany). For the detection of KSHV LANA, we used a rat monoclonal anti-LANA antibody reported previously [77]. The HRP-conjugated rabbit anti-rat IgG antibody (3050–05) was purchased from SouthernBiotech (Birmingham, USA), and the HRP-conjugated rabbit anti-mouse IgG (P0447) and goat anti-rabbit IgG (P0448) were purchased from Dako (Santa Clara, USA). The primary mouse antibody anti-ORF26 (clone 2F6B8) was purchased from LSBio (Seattle, USA) and the Cy-3 coupled secondary antibody (Cy3 AffiniPure Donkey Anti-Mouse IgG) from Jackson ImmunoResearch (Ely, Cambridgeshire, United Kingdom).

## Generation of KSHV-BAC16 mutants

To generate different mutations within the *pORF19* gene in the BAC16 carrying the KSHV genome, en passant mutagenesis was performed as described before [78]. Briefly, a kanamycin resistance cassette was amplified from a pOri6K.I-SceI vector with an integrated I-SceI cleavage site including homologous flanking sequences carrying the mutation by adding 0.2 μM of forward primer directly to a 25 μl reaction, followed by addition of the same amount of reverse primer after 17 PCR cycles. In an analogous way, the pORF19 knockout mutant was generated by replacing the pORF19 globular domain (residues 123 to 549) by the kanamycin resistance cassette. Subsequently, recombination proficient *E. coli* GS1783 carrying the KSHV BAC were transformed with the PCR amplicon by electroporation. The electroporated bacteria were selected for kanamycin and chloramphenicol resistance, and resistant clones were verified by restriction analysis.

For the second recombination step, kanamycin resistant clones were incubated with chloramphenicol only for 3 hours at 32°C, 1% L-arabinose was added for 1 hour at 32°C to induce I-SceI expression, and cultures were transferred to 42°C for 25 minutes to induce the recombination enzymes. After heat shock, cultures were returned to 32°C for 3 hours. Screening for loss of the kanR marker revealed final clones with kanamycin sensitivity.

The presence of introduced mutations in the mutants was confirmed by restriction analysis and Next Generation Sequencing of the entire KSHV BAC. For this purpose, purified BAC DNA was sheared by sonication. To avoid bias due to overamplification, library preparation was performed using the KAPA real-time library preparation kit (KAPA Biosystems, Wilmington, Massachusetts, USA) with a limited number of PCR cycles. Quality controlled libraries were sequenced on a MiSeq (Illumina, Berlin, Germany) using reagent kit v3 to generate $2 \times 300$ base paired-end reads. Reads were mapped to the KSHV BAC16 parental strain, and variants were identified by using the low frequency variant detector function in CLC genomics Workbench v9.

## Cell culture

HEK-293 (CRL-1573, American Type Culture Collection, ATCC) and the doxycycline inducible iSLK cells [46] (kindly provided by Frank Neipel, University Erlangen) were cultured in DMEM (Gibco, Thermo Fisher Scientific, Waltham, USA) supplemented with 10% fetal bovine serum (FBS, Sigma). For iSLK cells, 250 μg/ml G418 (Sigma, A1720) were added. Stable iSLK.BAC16.KSHV cell lines were generated by transfection of iSLK cells ($2 \times 10^5$ cells/6 well) with 2 μg BAC DNA of a Maxi preparation (Macherey-Nagel (Dueren, Germany), Nucleo-Bond BAC100) with Fugene 6 transfection reagent (Roche, 11 814 443 001) according to the manufacturer's protocol at a ratio of 3:1. The next day, cells were transferred from a 6-well plate into 10-cm dishes, and, another 24 hours later, cells were selected with 1.2 mg/ml Hygromycin B (PAN Biotech (Aidenbach, Germany), P06-08020) as described before [45] in the presence of G418. After 14 days, selected KSHV positive cell lines were maintained in the presence of 150 μg/ml Hygromycin B and G418.

## Lytic reactivation and titration of infectious progeny

KSHV lytic reactivation in the iSLK.BAC16.KSHV cell lines was induced by treating the cells with 2 mM sodium butyrate (SB) and 2 μg/ml doxycycline for 72 hours, leading to expression of the lytic switch protein RTA. To quantify the production of infectious viral particles, cell culture supernatants of induced iSLK cell lines were centrifuged for 5 minutes at 664 xg at 4˚C and added to $3 \times 10^4$ HEK-293 cells in a 96-well plate, which had been seeded the day before. The plate was centrifuged for 30 minutes at 32˚C and 450 xg and 72 hours postinfection, the cells were fixed with PFA and viral titers were calculated by counting GFP positive cells/well using a Citation 5 Cell Imaging Multi-Mode Reader from BioTek (Bad Friedrichshall, Germany).

## Cell lysis and western blot

After washing in PBS, cells were lysed in SDS lysis buffer (62.5 mM Tris-HCl pH 6.8, 2% (w/v) SDS, 10% (v/v) glycerol, 50 mM DTT, bromophenol blue). Samples were centrifuged for 10 minutes at 17.949 xg and 4˚C to pellet cell debris. If necessary, the lysates were sonicated for a few seconds before centrifugation. Protein concentrations were measured using a NanoDrop1000 (Peqlab, VWR, Darmstadt, Germany). The cleared cell lysates were boiled for 5 minutes at 95˚C and loaded on 8% to 12% SDS polyacrylamide gels. The Precision Plus Protein All Blue Prestained Protein Standards (1610373, Bio-Rad, Feldkirchen, Germany) was used as a protein marker. After SDS PAGE, the proteins were transferred on nitrocellulose membranes, unspecific binding was blocked by incubating the membranes in 5% milk in PBS-T buffer, and the primary antibody incubation was performed on a roller incubator at 4˚C overnight or 1 hour at room temperature (RT). After 3 washing steps in PBS-T, the membranes were incubated for 1 hour at RT with the secondary horseradish peroxidase (HRP)-conjugated antibody. To visualize the detection of specific proteins, the membranes were developed in a LAS-3000 Imager (Fujifilm, Duesseldorf, Germany) using either the SuperSignal West Femto Maximun Sensitivity Substrate (34096, Thermo Fisher Scientific, Waltham, USA) or self-made enhanced chemiluminescence (ECL).

## Purification of KSHV particles

iSLK cells transfected with wt or mutant pORF19 BAC constructs, their complemented counterparts and non-infected iSLK cells were assayed for virus production. Moreover, $4 \times 10^6$ cells were plated in a T150 Flask (10 flasks per each cell line) in a final volume of 20 ml per flask. Twenty-four hours later, KSHV lytic reactivation was induced by treating the cells with 2 mM SB (Merck) and 2 μg/ml doxycycline (Sigma, Taufkirchen, Germany). Forty-eight hours postinduction, the supernatants were collected, debris and cells were removed by centrifugation at 1,878 xg for 10 minutes at 4˚C, and the virus was pelleted 4 hours at 27,632 xg and 4˚C. The virus pellet was resupended overnight in 5 ml of DMEM, 10% FBS on ice, layered onto 2 ml 15% sucrose (w/v,) and centrifuged 1 hour at 72,000 xg at 4˚C. The resulting pellet was resuspended in 300 μl of MNT buffer (20 mM MES; 100 mM NaCl; 30 mM Tris; pH 7.4 adjusted with KOH), and small aliquots were digested with DNase (Roche) according to the manufacturer's recommendation. Aliquots were snap frozen in liquid nitrogen and stored at −80˚C until further analysis (western blot or qPCR).

## Complementation of iSLK BAC16-pORF19 mutants

pORF19 wt was cloned into a retroviral vector named pSRS.SF.pORF19.mCMV.RFP670pre capable of expressing a transgene driven by the strong Murine leukemia virus derived promoter (SF) and an RFP for flow cytometry analysis under control of a minimal CMV promoter cloned into the modified U3 region of the 3′ long-terminal repeat (LTR) [47,48]. A control vector

encoded the human Trim5α cDNA together with the RFP gene. VSVg pseudotyped viral vector particles were produced as previously described [47] except that transfection of the producing 293T cells was performed using the CAPHOS transfection kit (Sigma) following the manufacturer's protocol. Supernatants were harvested 48 hours posttransfection and stored at −80˚C.

iSLKs transfected with wt or mutant pORF19 BAC constructs were transduced with the supernatant using DMEM 10% FBS supplemented with 4 μg/ml of protamine sulfate, centrifuged 1 hour at 37˚C, 800 xg and incubated for 6 hours at 37˚C before media was changed to DMEM 10% FBS. Seventy-two hours postinfection, $1 \times 10^7$ cells were sorted based on fluorescence intensity in the APC channel as a measure for expression of RFP670 using FACSAria Fusion (Becton Dickinson, Heidelberg, Germany).

## Immunofluorescence of reactivated iSLK cells transfected with wt or mutant pORF19 BAC constructs

iSLK cells transfected with wt or mutant pORF19 BAC constructs, their complemented counterparts and non-infected iSLK cells were plated on glass coverslips ($4 \times 10^5$ cells per well of a 6-well plate). Twenty-four hours later, KSHV lytic cycle was induced using 2 mM SB (Merck) and 2 μg/ml doxycycline (Sigma). Forty-eight hours postreactivation, cells were washed once with PBS and fixed with 4% paraformaldehyde (PFA; Roth, Karlsruhe, Germany) for 20 minutes at RT. After fixation, the coverslips were washed 3 times with PBS, and cells were permeabilized with 0.2% Triton X-100 in PBS for 10 minutes at RT. Unspecific binding was blocked by incubating the cells in 10% FBS in PBS for 1 hour at 37˚C. The primary antibody targeting pORF26 (LSBio; dilution 1:200 in 10% FBS in PBS) was added for 1 hour at 37˚C, cells were washed 3 times with PBS, and the Cy-3 coupled secondary antibody (CyTM 3-conjugated AffiniPure Donkey Anti-Mouse IgG; dilution 1:200 in 10% FBS in PBS) mixed with 4 μg/μl 4′,6-diamidino-2-phenylindole (DAPI; Thermo Fisher Scientific, Waltham, USA) was added for 1 hour at 37˚C. Cells were washed twice with 10% FBS in PBS, twice with PBS, and finally rinsed 10× with ddH2O and mounted on slides with 5 μl of ProLongTM Glass Antifade Mountant (Invitrogen, Thermo Fisher Scientific, Waltham, USA). The slides were dried at RT overnight in the dark and images taken with a ZEISS 980 Airyscan microscope.

## Electron microscopy of reactivated iSLK cells transfected with wt or mutant pORF19 BAC constructs

iSLK cells transfected with wt or mutant pORF19 BAC constructs, their complemented counterparts, and non-infected iSLK cells were plated on glass coverslips ($4 \times 10^5$ cells per well of a 6-well plate). Twenty-four hours later, KSHV lytic cycle was induced using 2 mM SB (Merck) and 2 μg/ml doxycycline (Sigma). Forty-eight hours postinfection, cells were fixed with 2% glutaraldehyde and 2.5% formaldehyde in cacodylate buffer (130 mM $(CH_3)_2AsO_2H$, pH 7.4, 2 mM $CaCl_2$, 10 mM $MgCl_2$) for 1 hour at RT and contrasted with 1% (w/v) OsO4 in cacodylate buffer (165 mM $(CH_3)_2AsO_2H$, pH 7.4, 1.5% (w/v) $K_3[Fe(CN)_6]$) followed by 0.5% (w/v) uranyl acetate in 50% (v/v) ethanol overnight. The cells were embedded in plastic (29.19 g Epon 812, 12.66 g DDSA, 16.58 g MNA, 0.75 ml DMP30; Serva, Heidelberg, Germany), and 50-nm ultrathin sections were cut parallel to the substrate. Images were acquired with a Morgani transmission electron microscope (FEI, Eindhoven, the Netherlands) at 80 kV.

## Negative Stain EM of pentameric pORF19KCTDCC

A total of 5-μl oxidized pORF19$_{KCTD}$CC in SEC buffer at a concentration of 0.03 mg/ml was applied onto glow discharged carbon grids (ultra-thin carbon film on a copper support grid,

400 Mesh, Electron Microscopy Science, Hatfield, USA). After 2-minute incubation, the grid was washed twice with water and stained afterward with 2% (w/v) uranyl acetate for 30 seconds. Negatively stained EM grids were imaged on a Tecnai $G^2$ 20 microscope (FEI, Thermo Fisher Scientific, Waltham, USA) operated at 200 kV. Images were taken with an on-axis bottom-mount Eagle 4k Camera (FEI, Thermo Fisher Scientific, Waltham, USA). For 2D classification, approximately 12.000 particles were picked using WARP [79]. RELION 3.0.8 [80] was used to extract approximately 12.000 particles using a box size of 260 Å and to generate 2D class averages.

## Statistical analysis

Statistical tests were performed in GraphPad Prism v9 on data generated from biological replicates, with appropriate correction for multiple comparisons: $^{****}P < 0.0001$, $^{***}P < 0.001$, $^{**}P < 0.01$, and $^{*}P < 0.05$.

## Supporting information

**S1 Fig. Model of the KSHV capsid vertices.** Schematic view of the penton vertex **(A, B)** and the portal vertex **(C, D)**, viewed from above (top) and from the side (bottom) illustrating the main differences. For clarity, the constituents of CVSC and PVAT are colored in orange with different shades depicting pORF19 and pORF32, respectively, and the SCP pORF65 is shown as triangle in light and dark gray on hexons and pentons, respectively. CVSC, capsid vertex–specific component; KSHV, Kaposi's sarcoma-associated herpesvirus; PVAT, portal vertex–associated tegument; SCP, small capsid protein.
(TIF)

**S2 Fig. SEC analysis of recombinant proteins. (A–D)** Elution profiles from a Superdex 200 Increase 10/300 GL size exclusion column obtained upon purification of individual CTDs from HCMV pUL77 **(A)**, KSHV pORF19 **(B)**, and MuHV-68 pORF19 **(C)**. In all cases, the majority of the protein elutes at a volume presumably corresponding to a monomer. **(D)** Elution profile of oxidized pORF19$_{KCTD}$CC analyzed by SEC aligned to the profile of the non-oxidized wt pORF19$_{KCTD}$. The vertical red dotted lines mark the elution volume of the oxidized pORF19$_{KCTD}$CC and the non-oxidized wt pORF19$_{KCTD}$ **(B)**. The oxidized protein elutes at an earlier volume, underlining the difference in oligomeric state. The underlying data for all chromatograms can be found in S3 Data. CTD, carboxyl-terminal domain; HCMV, human cytomegalovirus; KSHV, Kaposi's sarcoma-associated herpesvirus; MuHV-68, murid gammaherpesvirus 68; SEC, size exclusion chromatography; wt, wild-type.
(TIF)

**S3 Fig. Structural alignment.** Structural alignment of the globular domains of HSV-1 pUL25 (PDB 2F5U), HCMV pUL77, KSHV pORF19, and MuHV-68 pORF19, obtained via a pairwise comparison with the Dali server [37]. Magenta or yellow background color indicates the SSE of the individual proteins as determined by the ENDscript server [81]; numbering and SSEs of HSV-1 pUL25 are shown above the alignment. Residues in red within black framed boxes are conserved across all 4 herpesvirus orthologs. Lowercase letters denote insertions relative to HSV-1 pUL25. Residues mutated in this study to block pentamerization are colored and framed in cyan and the positively charged residues in the funnel region of the pentameric pORF19$_{KCTD}$ in blue. HCMV, human cytomegalovirus; HSV, herpes simplex virus; KSHV, Kaposi's sarcoma-associated herpesvirus; MuHV-68, murid gammaherpesvirus 68; SSE, secondary structure element.
(TIF)

**S4 Fig. Electrostatic surface potential of pUL25$_{CTD}$ and its orthologs. (A+B)** View on the face of pUL25 previously described to contain a large number basic patches representing positive charges (left panel, PDB 2F5U) and the opposite face (right panel) compared with the charge distribution on the surface of pUL25 orthologs in the same orientation). The electrostatic potential is represented and calculated as for Fig 3. CTD, carboxyl-terminal domain. (TIF)

**S5 Fig. Crystal packing of the pentameric pORF19$_{KCTD}$ ring.** Crystalline arrangement of the pentameric pORF19$_{KCTD}$ rings in the $P2_12_12_1$ space group (top panel; 2 rings per AU), the $P6_322$ space group (middle panel; 1 ring per AU) and the $P2_1$ space group (bottom panel; 2 rings per AU) in the side view (left) and top view (right). The first pentameric ring in each AU is colored green, the second one (spacegroups $P2_12_12_1$ and $P2_1$) is colored red to illustrate the respective crystal packing environment. In all cases, the rings interact laterally to form layers that are densely stacked; however, the type of lateral interactions differs between individual crystal lattices, suggesting that a pentamer is the functional assembly unit. AU, asymmetric unit; CTD, carboxyl-terminal domain. (TIF)

**S6 Fig. 2D classification of pORF19$_{KCTDCC}$.** Negative stain EM images were used for single particle analysis and obtained 2D class averages are shown. CTD, carboxyl-terminal domain; EM, electron microscopy. (TIF)

**S1 Table. Nomenclature of proteins important for capsid assembly across herpesviruses.** (DOCX)

**S2 Table. Diffraction data collection and refinement statistics.** (DOCX)

**S3 Table. Missing loops in the ortholog structures.** (DOCX)

**S4 Table. Structural similarity across orthologs.** Statistics from a pairwise analysis of listed pUL25 orthologs using the DALI server [37]. For reference, Z scores below 2 are meaningless, whereas values of around 50 are obtained when structures of the same protein from 2 different crystal forms are compared (serving as reference for proteins of roughly the same size). rmsd is the root mean square deviation between Cα atoms (in Å). The third line in each box is "N/NT", where the number of aligned residues (N) is compared to the total residues in the alignment (NT). "%id" indicates % amino acid identity after the alignment. (DOCX)

**S5 Table. Efficiency of pORF19$_{KCTD}$ pentamerization in vitro.** Protein concentration was 127 μM in all cases. Percentages of "monomer" and "oligomer" were calculated by integration of the monomer and the oligomer peaks as indicated in the c(s) distributions obtained from the sedimentation profiles in Fig 4A, right panel. Since monomer and oligomer co-sediment in the faster moving boundary, the concentration of the oligomer is thereby overestimated and should not be taken as an absolute value, but as a measure for the ability of the different conditions to promote pORF19$_{KCTD}$ oligomerization. CTD, carboxyl-terminal domain. (DOCX)

**S6 Table. List of oligonucleotides used in this study.** (DOCX)

**S1 Data. Excel spreadsheet containing the underlying numerical values for Fig 4A and 4D.**
(XLSX)

**S2 Data. Excel spreadsheet containing the underlying numerical values for Figs 5B and 6B.**
(XLSX)

**S3 Data. Excel spreadsheet containing the underlying numerical values for S2A–S2D Fig.**
(XLSX)

**S1 Raw Images. Original western blot images shown in Figs 5A, 6A and 6C.** The "X" designate samples not shown within the main text figure. The triplicates shown in the original blots for Fig 5A were analyzed for the purpose of Fig 5B.
(PDF)

## Acknowledgments

We thank Stephane Roche (I2BC, Gif-sur-Yvette), Kay Grünewald (CSSB, Hamburg), and members of the Krey lab for helpful discussions; Simon Krooss for help with statistics; staff of synchrotron beamlines Proxima-1 and -2 at SOLEIL for help during data collection; Lidia Litz for excellent technical assistance; Dr. Norbert Mücke (DKFZ, Heidelberg) for experimental determination of buffer density and viscosity; and Marija Backovic and Janna Bigalke for critical reading of the manuscript.

## Author Contributions

**Conceptualization:** Ute Curth, Luisa J. Ströh, Jens Bohne, Rudolf Bauerfeind, Beate Sodeik, Thomas F. Schulz, Thomas Krey.

**Data curation:** Ute Curth, Pierre Legrand, Thomas Krey.

**Formal analysis:** Peter Naniima, Eleonora Naimo, Sandra Koch, Ute Curth, Khaled R. Alkharsah, Luisa J. Ströh, Eva Maria Borst, Jens Bohne, Martin Messerle, Rudolf Bauerfeind, Pierre Legrand, Beate Sodeik, Thomas F. Schulz, Thomas Krey.

**Funding acquisition:** Thomas Krey.

**Investigation:** Peter Naniima, Eleonora Naimo, Sandra Koch, Ute Curth, Khaled R. Alkharsah, Luisa J. Ströh, Anne Binz, Jan-Marc Beneke, Benjamin Vollmer, Heike Böning, Rudolf Bauerfeind, Pierre Legrand, Thomas Krey.

**Methodology:** Peter Naniima, Eleonora Naimo, Sandra Koch, Ute Curth, Khaled R. Alkharsah, Luisa J. Ströh, Anne Binz, Heike Böning, Jens Bohne, Rudolf Bauerfeind, Pierre Legrand, Thomas Krey.

**Project administration:** Thomas Krey.

**Resources:** Prashant Desai.

**Supervision:** Jens Bohne, Beate Sodeik, Thomas F. Schulz, Thomas Krey.

**Validation:** Thomas Krey.

**Visualization:** Peter Naniima, Eleonora Naimo, Ute Curth, Luisa J. Ströh, Thomas F. Schulz, Thomas Krey.

**Writing – original draft:** Ute Curth, Beate Sodeik, Thomas F. Schulz, Thomas Krey.

**Writing – review & editing:** Peter Naniima, Eleonora Naimo, Sandra Koch, Ute Curth, Khaled R. Alkharsah, Luisa J. Ströh, Benjamin Vollmer, Eva Maria Borst, Jens Bohne, Martin Messerle, Rudolf Bauerfeind, Pierre Legrand, Beate Sodeik, Thomas F. Schulz, Thomas Krey.

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
