## [Editor Report · Decision Letter 0]

27 Jan 2021

Dear Dr. Krey, 

Thank you for submitting your manuscript entitled "Assembly of infectious Kaposi’s sarcoma-associated herpesvirus (KSHV) progeny requires formation of a pentameric pORF19 portal cap" for consideration as a Research Article by PLOS Biology.

Your manuscript has now been evaluated by the PLOS Biology editorial staff, as well as by an academic editor with relevant expertise, and I am writing to let you know that we would like to send your submission out for external peer review.

Please re-submit your manuscript within two working days, i.e. by Jan 29 2021 11:59PM.

Kind regards,

Paula

---

Associate Editor

PLOS Biology

---

## [Decision Letter · Decision Letter 1]

10 Mar 2021

Dear Dr. Krey,

Thank you very much for submitting your manuscript "Assembly of infectious Kaposi’s sarcoma-associated herpesvirus (KSHV) progeny requires formation of a pentameric pORF19 portal cap" for consideration as a Research Article at PLOS Biology. Your manuscript has been evaluated by the PLOS Biology editors, an Academic Editor with relevant expertise, and by several independent reviewers.

In light of the reviews (below), we will not be able to accept the current version of the manuscript, but we would welcome re-submission of a much-revised version that takes into account the reviewers' comments. We cannot make any decision about publication until we have seen the revised manuscript and your response to the reviewers' comments. Your revised manuscript is also likely to be sent for further evaluation by the reviewers.

In particular, reviewer #1 suggests you to add a schematic picture to describe the penton components, recommends to do a 2D classification to demonstrate that the protein oligomerize forming a pentamer, asks you why did you filtered the simulated map at 5 Å, and points out that you didn't submit the PDB validation reports for their X-ray structures. Reviewer #2 has concerns regarding the overinterpretation of the biochemical and the virological data, the inadequate description of experimental procedures, and the missing controls. Reviewer #2 thinks that you need to show that the portal indeed contains pORF19 or tone down your conclusion that the pentameric pORF19 structure presented in the manuscript is present in the portal cap (including abstract and title). This reviewer also says that you need to add controls to demonstrate that pORF19 is essential for capsid assembly, points to discrepancies in the mutant phenotypes and says that you need proper controls to resolve them, and says that some experimental procedures need to be described with sufficient detail and show all pieces of data. Finally, reviewer #2 also says that you should discuss a substantial body of literature on the role of HSV-1 UL25 in capsid stability and DNA packaging.

We expect to receive your revised manuscript within 3 months. 

**IMPORTANT - SUBMITTING YOUR REVISION**

*Re-submission Checklist*

*Published Peer Review*

*PLOS Data Policy*

*Blot and Gel Data Policy*

Sincerely,

Paula

---

Associate Editor,

pjaureguionieva@plos.org,

PLOS Biology

REVIEWS:

Reviewer #1: Structural Virology

Reviewer #2: Structural biology and herpesvirus

Reviewer #1: The article by Naniima et al. describes not only the crystal structure of pentameric pORF19 of KSHV (and its β-herpesviral ortholog) but it also integrates a series of mutagenesis, cellular, fluorescence and TEM microscopy experiments that shed light on the portal cap structure and its functional role in the herpesvirus life cycle. The findings are of interest and novel to the virology community.

Overall, the manuscript is a well presented, consistent piece of work and should be published in PLoS Biology after the suggested revisions are considered and properly addressed. 

Here below some comments that would possibly help the clarity of the text.

ABSTRACT

Ok

INTRODUCTION

Page 1, 8 lines from top: why 'pseudo-icosahedral capsid'? better to use 'icosahedral capsid' 

Page 2: the description of the penton components would benefit from a schematic picture for those who are not expert in the herpesvirus field, serving as road map of the different CVSC (or CAT), triplex proteins etc. on top of a schematic icosahedron. It can be a schematic figure as supplementary Fig. S1 (this would also help the interpretability of Figure 3 in the context of the virion structure).

RESULTS:

Page 6, 2 lines from below: the sentence 'These data suggest…..significance' should be omitted because it does not represent an experimental result but the possible interpretative scenarios of the Figure S2. For this reason, it is better to move it to the Discussion section. 

Page 7, 11 lines from below: please remove the second decimal to 19.22 kcal /mol also because the symbol '~' is used. 

Page 8: The text referring to Figure 3 and Figure 3 is confusing. Figure 3 labels are incorrectly placed; moreover Figure 3 can definitely be improved in terms of visual clarity by linking different regions with larger insets. Please clarify the main-text, the Figure 3 and its corresponding caption.

Page 11, 4 lines from top: the sentence 'negative-stain EM of pORF19KCTDCC demonstrated a 5-fold symmetry' is overstated. To demonstrate that the protein oligomerize forming a pentamer and thus displaying a 5-fold symmetry, a 2D classification would be more appropriated - and it is indeed recommended. Using negative-stain technique the Authors would not require a large set of images to prove their statement. 

Figure 4H, as it is now, is not convincing and it does not show the existence of an homogeneous population of pORF19KCTDCC that forms a stable pentamer.

DISCUSSION:

Page 19: 12 lines from top: the sentence 'The recombinant portal cap…..capsid disassembly'. It is not clear why the Authors mention this as it seems more of a plan that will possibly happen in the future. It does not add much to the discussion.

FIGURES:

Figure 3: see above comments

Figure 4H: see above comments

Figure 7 caption: where is it the scale bar? Also, are these Z sections or stacked images?

METHODOLOGY:

Page 24: 'Structure analysis': The Author should state why the simulated map from the crystal structure was set to 5 Å resolution for the fitting into the portal density. The claimed resolution in the EMD-20430 is 7.6 Å for the asymmetric 3D reconstruction and it is likely that the resolution of the portal cap is slightly lower so why has the Author filtered the simulated map at 5 Å and not 7.6 or 8 Å? 

The authors didn't submit the PDB validation reports for their X-ray structures. In this case Table S2 is pretty convincing but it would a good practice for authors describing X-ray structures to include in their submission the corresponding PDB validation report: https://www.wwpdb.org/validation/validation-reports

Reviewer #2: Summary

Herpesviral capsids are complex icosahedral assemblies composed of many copies of several viral proteins. One of these, the outer capsid protein conserved among all known herpesviruses -termed UL25 in HSV-1 - reinforces the capsid to allow it to withstand the pressure of the packaged genome and has also been implicated in genome packaging, cleavage, and retention. Prior to this study, the structural information has only been available for the HSV-1 UL25 homolog. Here, Naniima et al present the structures of three additional homologs of UL25: KSHV pORF19, MHV68 pORF19, and HCMV UL77. Interestingly, the KSHV pORF19 crystallized as a pentamer. By fitting the pentameric crystal assembly into the cryo-EM densities for the portal cap on the KSHV capsid, the authors conclude that the pORF19 pentamer observed in the crystals represents the portal cap. Mutations at the pentameric interface reduce viral titer and abolish capsid assembly, prompting the conclusion that formation of the pORF19 pentamer is essential for capsid assembly. The structural data are solid, but there are concerns regarding the overinterpretation of the biochemical and the virological data, the inadequate description of experimental procedures, and the missing controls.

Major criticisms

1. Cryo-EM reconstructions of HSV-1 and KSHV capsids (Gong 2019 and Liu 2019) have revealed the presence of the portal cap and suggested that it contains UL25. However, the presence of UL25 within the portal cap has not yet been shown conclusively. The authors reference both Newcomb 2001 and Trus 2004 in the introduction (refs 9 and 10) as studies that agree with these proposed density assignments in the EM studies, yet both references focus on the HSV-1 portal composed of UL6 and do not examine the presence or involvement of UL25 at the portal cap. To be able to conclude that pORF19, a UL25 homolog, is a component of the portal cap (and that the pentameric structure reported here could be important for its formation), the authors need to show that the portal, indeed, contains pORF19, e.g., by immunogold labeling, or, alternatively, show that the deletion of pORF19 eliminates the portal cap from the capsids. Short of this, the conclusion that the pentameric pORF19 structure presented here is present in the portal cap should be toned down throughout the manuscript (including the title and the abstract) and presented as a hypothesis rather than a conclusion.

2. The observation that mutations at the pentameric interfaces in pORF19 abolish capsid assembly in infected cells contradict the author's own data that show capsid assembly even in the absence of pORF19 in insect cells expressing capsid proteins. A bigger concern is the fact that this finding contradicts a body of literature on capsid assembly in alphaherpesviruses, such as HSV-1 and PRV, that shows that A- and B-capsids can form in infected cells even in the absence of UL25 (McNab 1998, Stow 2001, Mettenleiter 2006, Baines 2014). For example, C-terminal truncations in HSV-1 UL25 do not preclude formation of A- or B-capsids (Baines 2014). Furthermore, KSHV capsids can form in vitro the absence of the portal (Grzesik 2017), a finding confirmed by the authors. One potential explanation is that the mutant viruses contain additional, unintended mutations that disrupt capsid assembly. This interpretation is consistent with poor in-trans complementation with WT pORF19. Alternatively, UL25 homologs could have differing roles in capsid formation during infection such that whereas HSV-1 UL25 is not essential for capsid formation, its KSHV homolog pORF19 is. This finding would be very interesting but requires that the authors perform additional controls, namely, a positive control showing that capsids do not form in the absence of pORF19 as well as a negative control showing that mutations located away from the pentameric interface has no effect on capsid formation.

Along the same lines, it is difficult to reconcile the observed differences in the mutant phenotypes observed in Fig. 6 vs. Figs. 7 and 8. For example, the DQ and VL mutants appear to have a similar defect in progeny production (Fig. 6). However, only DQ appeared to form capsids (Fig. 7), predominantly A- and B-capsids, whereas the VL mutant produced no capsids (as observed in Figs, 7 and 8). Proper controls are needed to resolve these discrepancies. 

3. There is a concern about the experimental rigor because some experimental procedures are not described in sufficient detail and because a few pieces of data are not shown. First, authors must remedy the lack of experimental details throughout the entire manuscript. For example, the description of the procedure used to test capsid association of the pentameric interface mutants is scant. The authors need to describe their experimental procedure in sufficient detail to allow others to reproduce their experiments. Citing the original paper (Desai, 2017) is insufficient. As another example, there is no mentioning in the methods of how the cysteine mutants were oxidized. Second, it is essential that the authors show all of their control data. For example, the data for the KO mutant of pORF19 should be shown to establish the validity of the conclusions. The authors should also clarify what KO mutation is, i.e., which amino acid residues are deleted. 

4. Describing pORF19 as a portal cap protein ignores its other functions and is not sufficiently supported by the available data. The authors should discuss a substantial body of literature on the role of HSV-1 UL25 in capsid stability and DNA packaging, e.g., Snijder 2019 and Cockrell 2009, as it would permit comparisons and contrasts with gammaherpesviruses. The excessive description of capsid structure in the already lengthy introduction could be shortened to accommodate these changes. 

Additional concerns

5. The manuscript is missing both the page numbers and line numbers.

6. Figure 3 is confusing and needs to be revised to be more helpful to the reader. The authors are trying to show that the pORF19 pentamer does not fit into the penton CVSC/CATC densities, yet the cartoon models shown in Fig. 3D are not the pentameric form that was obtained in the crystals. It would be helpful to show the fit of the pORF19 pentamer into the portal cap (as already shown in Fig. 3B) side-by-side with its fit into the CVSC/CATC density and include measurements showing that the pentamer does not fit the CVSC/CATC pORF19 density (one can use different colors to indicate where pORF19 is in the CVSC/CATC). In addition, the labels on Fig. 3 are incorrect (Fig. 3AB shows the portal, but in the text is referred to as the CVSC/CATC).

7. The authors present three new structures of UL25 homologs yet do not compare/contrast them in any meaningful way in Figure 1. It would be helpful to demonstrate structural similarities and differences using structural alignments, either all four aligned together or pairwise. It would also be helpful to have the sequence alignments indicating experimentally derived secondary structure for all four homologs. Comparisons of the electrostatic surface potential should also be shown in the main figure, instead of the supplement.

8. The pORF19 pentamer forms a tunnel lined with positively charged residues. The authors should discuss whether this is a common feature among the homologs or is specific to KSHV pORF19. Do homology models of the pentamers for the other homologs also show a positively charged tunnel? If not, what implications does this have regarding its role? 

9. Fig. 2 should show the measurements listed in the Results section.

10. The sentence "Based on cryo-EM data and a pORF19 homology model a loop…" is confusing as written and should be rephrased. 

11. The cysteine mutants (P137C and P461C) are labeled incorrectly in Fig. 4F (currently labeled as P16C).

12. Figure 4 is hard to follow. For example, panel F is located between G and H. It would be helpful if the authors moved the SEC panels to the top of the figure, from right to left (SEC buffer, LiAc buffer, then mutant chromatograms). Directly underneath, in a second row, could be panels B, F, H (in the text it is currently 3F, G and H discussed before C, D, and E). The third row would be C, E, D (see comment below) with the mutant information.

13. To help the reader, the three mutants - DQ, loop, and VL - should be listed in the same order both in the text and the figures.

14. In the Results section describing the association of ORF19 mutants to capsid vertices, the authors state "In all cases, density gradient centrifugation of cell lysates revealed a clear capsid band at the expected position." What position is being referred to? It would be helpful to indicate this in the figure legend or, better, show the gradients themselves.

15. Molecular masses of proteins in Fig. 5 and 6 should be indicated.

16. Fig. 7 legend refers to scale bars, yet none are shown on the figure. 

17. Primers use in cloning should be provided as a Supplemental Table, for transparency.

18. The strain name of E. coli for the KSHV BAC is incorrect as listed. GS1783 corresponds to HSV-1 BAC.

19. In the methods, FBS is defined once, but further in the text FCS is used and is not defined.

---

## [Decision Letter · Decision Letter 2]

10 Aug 2021

Dear Dr Krey,

Thank you for submitting your revised Research Article entitled "Assembly of infectious Kaposi’s sarcoma-associated herpesvirus (KSHV) progeny requires formation of a pORF19 pentamer" for publication in PLOS Biology. I have now obtained advice from the original reviewers and have discussed their comments with the Academic Editor. 

Based on the reviews, we will probably accept this manuscript for publication, provided you satisfactorily address the remaining points raised by the reviewers. Please also make sure to address the following data and other policy-related requests.

Reviewer #2 says that their finding that the pORF19 pentamer crystal structure fits into the density of the portal cap in the KSHV capsid cryoEM reconstruction, suggesting that the crystal structure determined herein represents the portal cap is not yet supported by any biochemical or functional data. Therefore, you should tone down this section of the discussion. You should also add a thoughtful discussion on how pORF19 contributes to assembly and how this compares to pORF19 homolog in HSV, and a thorough discussion of the role of pORF19 in KSHV capsid assembly and how this may be different in other herpesviruses, especially HSV. Reviewer #2 also asks you to rule out the possibility that pORF19 contributes to capsid assembly indirectly. We consider that it would be enough if you discuss different potential mechanisms by which pORF19 could contribute to assembly in the Discussion. Furthermore, reviewer #2 also thinks that you should discuss further the hypothesis that capsid assembly in insect cells can occur even in the absence of pORF19 because the capsid proteins are present at much higher levels in the KSHV-infected cells, and clarify Fig. 4D.

We suggest that you delete the acronym from the title in order to make it easier to read: "Assembly of infectious Kaposi's sarcoma-associated herpesvirus progeny requires formation of a pORF19 pentamer". 

DATA POLICY:

Regardless of the method selected, please ensure that you provide the individual numerical values that underlie the summary data displayed in the following figure panels as they are essential for readers to assess your analysis and to reproduce it: Figures 4AD, 5B, 6B, S2ABCD.

**Please also ensure that figure legends in your manuscript include information on where the underlying data can be found, and ensure your supplemental data file/s has a legend.**

We require the original, uncropped and minimally adjusted images supporting all blot and gel results reported in an article's figures or Supporting Information files. We will require these files before a manuscript can be accepted so please prepare and upload them now. Please carefully read our guidelines for how to prepare and upload this data: https://journals.plos.org/plosbiology/s/figures#loc-blot-and-gel-reporting-requirements 

We need this for figures 5A, 6AC.

We expect to receive your revised manuscript within two weeks.

*Published Peer Review History*

*Early Version*

Sincerely,

Paula

---

Associate Editor,

pjaureguionieva@plos.org,

PLOS Biology

Reviewer remarks:

Reviewer #2: 1. There are two major findings in this manuscript. The first finding is that the pORF19 pentamer crystal structure fits into the density of the portal cap in the KSHV capsid cryoEM reconstruction, suggesting that the crystal structure determined herein represents the portal cap. The second finding is that the capsids do not assemble in the absence of pORF19 or when the pORF19 pentameric interfaces are mutated.

The former finding is interesting but is not yet supported by any biochemical or functional data. So, its biological relevance is yet unclear. Nonetheless, the authors focus the entirety of their discussion on how the pORF19 pentamer could form at the portal cap (lines 502-541) and how their data suggest there is likely a UL77 portal cap in HCMV (lines 491-501). Considering the unclear biological relevance of the pentameric structure reported here, the authors could tone down this section of the discussion.

By contrast, the latter finding is more interesting (it is even reflected in the title) and, potentially, more biologically significant because while pORF19 appears to play an essential role in capsid assembly in gammaherpesviruses, the HSV-1 UL25 homolog is dispensable for capsid formation (McNab 1998, Stow 2001, Mettenleiter 2006, Baines 2014). If true, this may suggest major differences in the assembly mechanisms of herpesviruses from different subfamilies. However, a thoughtful discussion on how pORF19 contributes to assembly and how this compares to pORF19 homolog in HSV is still lacking despite this being a major, biologically significant claim of the manuscript that is highlighted in the title. The importance of the pentamer for assembly is only mentioned briefly in the Results section (line 450-453). This gap was highlighted in the original criticism (see Reviewer 2 comment #4) but has not been sufficiently addressed in the revised manuscript. A thorough discussion of the role of pORF19 in KSHV capsid assembly and how this may be different in other herpesviruses, especially HSV, is warranted and would boost the impact of the manuscript. This should be done in the Discussion section.

2. It is important that the authors rule out the possibility that pORF19 contributes to capsid assembly indirectly, for example, by being important for capsid protein expression (see, for example, a study on the role of the small capsid protein in KSHV capsid assembly - Sathish and Yuan 2010). The authors show that in the absence of pORF19, less pORF26 (triplex) is detected, and suggest that this could be due to reduced pORF26 expression (lines 410-412) but stop short of testing it. Ideally, the expression of other capsid proteins, such as the major capsid protein, should also be tested. This is a relatively straightforward experiments that would support a major (albeit undersold) claim of the manuscript. This would boost the claim that pORF19 contributes to assembly directly. At the bare minimum, the authors should discuss different potential mechanisms by which pORF19 could contribute to assembly in the Discussion. Is it known that the anti-pORF26 antibody is specific to capsid-bound pORF26? Knowing this would help with the data interpretation.

3. The observation that mutations at the pentameric interfaces in pORF19 abolish capsid assembly in infected cells contradict the data that show capsid assembly even in the absence of pORF19 in insect cells expressing capsid proteins. The authors now hypothesize that capsid assembly in insect cells can occur even in the absence of pORF19 because the capsid proteins are present at much higher levels in the KSHV-infected cells. But this is mentioned only briefly (lines 429-430) and should be discussed further, especially considering that the in vitro HSV-1 capsid assembly system largely recapitulates what is seen in HSV-1-infected cells (reviewed in Homa and Brown 1997, Brown and Newcomb 2011, and Heming 2017). While the authors agree that "These data clearly indicate functional differences between HSV-1 pUL25 and KSHV pORF19…", these differences are not compared in enough detail throughout the text. What is the evidence that the capsid protein levels are, indeed, much higher in insect cell system compared to KSHV-infected cells?

Minor criticism:

4. Fig 4D is still confusing. The authors do not explain why the Y axis was normalized nor how it is normalized. The authors should include pORF19KCTD in the same panel and indicate LiAC buffer. The difference between the oligomer of pORF19KCTD and pORF19KCTDCC ox should also be explained.

---

## [Editor Report · Decision Letter 3]

22 Sep 2021

Dear Dr Krey,

On behalf of my colleague Paula Jauregui (who is currently out of the office) and the Academic Editor, Bill Sugden, I am pleased to say that we can accept your study "Assembly of infectious Kaposi’s sarcoma-associated herpesvirus progeny requires formation of a pORF19 pentamer" in principle for publication in PLOS Biology, provided you address any remaining formatting and reporting issues. These will be detailed in an email that will follow this letter and that you will usually receive within 2-3 business days, during which time no action is required from you. Please note that we will not be able to formally accept your manuscript and schedule it for publication until you have made any required changes.

Thank you for having chosen PLOS Biology as a venue for the publication of your work supporting Open Access publishing. We look forward to publishing your study. 

PRESS

With best wishes,

Nonia

Nonia Pariente, PhD

Editor in Chief

PLOS Biology

on behalf of  

Paula Jauregui, PhD 

Senior Editor 

PLOS Biology
